# RNA Structure: Past, Future, and Gene Therapy Applications

**DOI:** 10.3390/ijms26010110

**Published:** 2024-12-26

**Authors:** William A. Haseltine, Kim Hazel, Roberto Patarca

**Affiliations:** 1ACCESS Health International, 384 West Lane, Ridgefield, CT 06877, USA; kim.hazel@accessh.org (K.H.); roberto.patarca@accessh.org (R.P.); 2Feinstein Institutes for Medical Research, 350 Community Dr., Manhasset, NY 11030, USA

**Keywords:** RNA structure, tertiary structure, helix structure, gene therapy, messenger RNA, small interfering RNA, long non-coding RNA

## Abstract

First believed to be a simple intermediary between the information encoded in deoxyribonucleic acid and that functionally displayed in proteins, ribonucleic acid (RNA) is now known to have many functions through its abundance and intricate, ubiquitous, diverse, and dynamic structure. About 70–90% of the human genome is transcribed into protein-coding and noncoding RNAs as main determinants along with regulatory sequences of cellular to populational biological diversity. From the nucleotide sequence or primary structure, through Watson–Crick pairing self-folding or secondary structure, to compaction via longer distance Watson–Crick and non-Watson–Crick interactions or tertiary structure, and interactions with RNA or other biopolymers or quaternary structure, or with metabolites and biomolecules or quinary structure, RNA structure plays a critical role in RNA’s lifecycle from transcription to decay and many cellular processes. In contrast to the success of 3-dimensional protein structure prediction using AlphaFold, RNA tertiary and beyond structures prediction remains challenging. However, approaches involving machine learning and artificial intelligence, sequencing of RNA and its modifications, and structural analyses at the single-cell and intact tissue levels, among others, provide an optimistic outlook for the continued development and refinement of RNA-based applications. Here, we highlight those in gene therapy.

## 1. RNA as an Organic Code

The seven-note musical alphabet played with variations in number, tempo, intensity, rhythm, pitch, or instrument, has produced myriads of musical melodies over millennia. Similarly, the organic codes of nucleic acid, protein, polysaccharide, and lipid biomolecules [1,2,3] use relatively few symbols to generate vast information-storing combinations with context-dependent meanings, underlying the diversity and complexity of Earthly life [3,4,5]. Underscoring the analogy between the human-made music code and organic codes, sonification tools provide auditory displays of individual and collective biomolecule sequence information as an adjunct to visual and analytical bioinformatics tools [6,7,8]. Understanding organic codes is fundamental to therapeutics’ development.

The hypothesis that the emergence of overlapping organic codes heralded the living organisms (biotic) era is in line with the intricate interdependence between deoxyribonucleic acid (DNA), ribonucleic acid (RNA), proteins, carbohydrates, lipids, and metabolites in cell- and capsid (viruses)-based organisms [9,10,11,12].

The genetic code is the core of life [13], and DNA is its blueprint [14]. The 1800s saw crucial developments in this field. Justus Liebig reported an acidic material in a beef muscle filtrate. Friedrich Miescher discovered ‘nuclein’ in leukocyte nuclei as a protein-degradation-resistant, phosphorus-rich, natural living system’s chemical and chromosome structural component [15]. Richard Altmann coined the term nucleic acid [16].

By 1910, two kinds of nucleic acids were distinguished based on sources and isolation methods: the thymonucleic or zoonucleic acid, now termed DNA, from thymus or animals, and the phytonucleic acid, now termed RNA, from yeast and plants [17,18]. Later, both were found ubiquitously in living organisms. Levene and collaborators [19,20,21] identified the planar aromatic ring structures of DNA’s constituent purine (adenine [A] and guanine [G]) and pyrimidine (thymine [T] and cytosine [C]) nitrogenous bases, and that RNA has the pyrimidine uracil [U], instead of thymine, and pentose instead of hexose as carbohydrate (Figure 1).

In 1944, Oswald Avery [22] proposed that DNA is the genetic information carrier. In 1950, Erwin Chargaff [23] deciphered the consistent proportions of DNA’s constituent bases and the A-T and C-G base pairing rules. In 1953, Maurice Wilkins [24,25] and Rosalind Franklin with Raymond Gosling [26,27] conducted the X-ray diffraction and crystallography studies that led James Watson and Francis Crick to discover the double-helical structure of DNA in 1953 [28,29,30] as the foundation for the DNA theory of inheritance [31]. Each nucleotide interacts with water, ions, amino acids, small molecules, and every other nucleotide, stabilizing the structure [32].

The Watson–Crick right-handed helical B-DNA is the native form of DNA in cells. However, DNA’s helical structure and biological properties can vary transiently along short repetitive tracts, as in left-handed Z-DNA [33,34,35,36], or reversibly en masse, as in the transition between B- and A-DNA in microorganisms in extreme temperatures and pH [37,38]. Other higher-order variations include supercoils (double helix ends join in bacterial genomes), bubbles, hairpins and cruciforms (when palindromes are present), slipped loops, three-stranded triple helices (H-DNA), and tetrameric i-motifs (over 50,000 in the human genome) and related four-stranded G-quadruplexes [39,40,41]. Tertiary DNA structures vary from person to person in critical genes like the insulin gene, constituting therapeutic targets [42].

First proposed by Mitsui et al. in 1970 [43] and later proven by Wang AH et al. [33], Z-DNA is a left-handed helix in equilibrium with the lower energy right-handed B-DNA. Flipons, typically involving an alternating purine/pyrimidine motif, can flip between B- and Z-DNA conformations under physiological conditions aided by binding proteins, introducing diversity to transcriptomes, particularly in immunity and transcription functions [35,36,44].

Discovered by Franklin and Gosling in 1953 [45] in DNA crystals after dehydration, A-DNA, also derived from protein binding to DNA, is a right-handed double helix but with a shorter and more compact helical structure than B-DNA, resulting in slightly more base pairs per turn, a smaller twist angle, and a shorter rise per base pair. The major groove of A-DNA is deep and narrow, the minor groove is wide and shallow, and the base pairs are not perpendicular to the helix-axis as in B-DNA. A-DNA can occur in DNA-RNA hybrid double helices and double-stranded RNAs. RNA can only form an A-type double helix because of the steric restrictions imposed on ribose by the 2′ hydroxyl residue [40].

After Z-DNA was discovered and named after its sugar-phosphate backbone’s zig-zag course as an alternative to the more common Watson–Crick B-DNA, nuclear magnetic resonance and other studies showed that the common A-RNAs, particularly those with higher Guanine/Cytosine content, could similarly undergo the right-to-left-handed conformational change to the higher energy Z-RNA [44,46,47,48]. Z-binding proteins specifically recognize and bind Z-DNA [49,50,51] and Z-RNA [52,53]. Z-DNA and Z-RNA encoded by flipons under physiological conditions are implicated in various biological processes, including transcription and immunity [44]. Z-RNA has been studied less than Z-DNA, and both are challenging to detect in vivo.

DNA organizes in the cellular nucleus into nucleosomes. Nucleosomes are the basic units of eukaryotic chromatin, the 3D structure of tightly folded chromosomes to fit into cellular nuclei. Nucleosomes are formed by 147 bp of duplex DNA wrapped around an octamer of histones [54], followed by a linker DNA bound by histone (H1) in complex eukaryotes [55]. Nucleosomes reduce access to >95% of the DNA [56], and maintain a defined architecture along the genome, with certain positions with well-positioned nucleosomes [57,58,59,60,61,62,63]. The most significant nucleosome-free regions are associated with gene promoter regions upstream of the transcription start sites, replication origins, and transcription termination sites [64,65]. The widths of nucleosome-free regions correlate with gene expression [66]. Nucleosome architecture perturbation associated with stress, cell cycle phase changes, nutrient sources, or the cell metabolic cycle underscores the association between nucleosome architecture and gene activity [60,64,67,68].

Structural similarities between RNA and DNA (Figure 1) allow the formation of RNA-DNA hybrids, such as the R loops, which also include a displaced single-stranded DNA [69]. Antisense noncoding RNAs may form R lops. R loops accumulate throughout the genome in pericentromeric DNA, telomeres, ribosomal DNA, or transcription termination among other regions, and are involved in transcription and chromatin structure. Because they can also adversely affect genome stability and replication, several DNA and RNA metabolism factors, such as ribonucleases, RNA-DNA helicases, RNA processing factors, and topoisomerase I, degrade R-loops or prevent their formation [69].

As an example of a protein interacting with DNA and RNA, the topoisomerase I enzyme prevents genomic instability by alleviating DNA torsional strain. Topoisomerase I introduces transient single-strand breaks that prevent the accumulation of supercoiling and torsional stress, which could otherwise lead to damage and instability of DNA and cell death [70]. Interactions between RNA and Topoisomerase I regulate DNA during transcription by modulating Topoisomerase I-mediated relaxation. In cancer cells, for instance, DNA transcription is often elevated, necessitating increased levels of Topoisomerase I activity to relax the DNA and maintain proper gene expression. RNA opposes Topoisomerase I activity. Inhibiting RNA binding of Topoisomerase I may work similarly to antineoplastic Topoisomerase I inhibitors like camptothecin by increasing Topoisomerase I catalytic complexes on DNA [70]. Therefore, R-loops and interacting proteins are therapeutic targets.

Beyond cancer, dysfunction of R loop-interacting factors in several genetic diseases leads to replication stress, genome instability, chromatin alterations, or gene silencing [69]. Furthermore, many chromatin-associated complexes, including histone modifiers, transcription factors, and DNA methyltransferase, interact with RNA [71]. RNA can also promote the repair of double-strand breaks in DNA by helping position and holding the broken DNA ends in place and guiding the cellular repair machinery, thereby contributing to genome integrity [72].

## 2. RNA Has Many Functions Through Its Intricate, Ubiquitous, Diverse, and Dynamic Structure

RNA has emerged as a central biomolecule in the multidirectional flow of genetic information for phenotype and biological diversity generation [12,73]. RNA is no longer considered simply an intermediary between the data stored in DNA and that functionally displayed in proteins. Although about 70–90% of the human [74] and 85–90% of the yeast genome [75] are transcribed into RNA, much remains unknown about RNA functions in cells [76]. This has fueled the rise in prominence of RNA-based therapeutics.

RNA structure plays critical roles in every step of RNA’s lifecycle, including transcription, splicing, localization [77,78], translation [79,80], and RNA decay [81]. However, RNA structure differs among individual cells and provides additional information in defining cellular identities by, for instance, informing RNA-binding protein binding and gene regulation [82]. To this end, overall RNA structure profiles better discriminate cell type identity and differentiation stage than gene expression profiles alone. For instance, RNA structure is more homogeneous in human embryonic stem cells than differentiating neurons, with the highest homogeneity in coding regions. More extensive heterogeneity is found within 3′ untranslated regions and is determined by specific RNA-binding proteins. Moreover, the cell-type variable region of 18S ribosomal RNA is associated with cell cycle and translation control. It is, therefore, important to systematically characterize RNA structure-function relationships at single-cell resolution using approaches such as single-cell structure probing of RNA transcripts [82]. To this end, the sc-SPORT high-throughput approach allows the study of RNA structures in single cells by optimizing conditions that increase mutation rates and efficiencies of library preparation and second-strand synthesis. Although the approach also includes a computational pipeline to analyze heterogeneous RNA structures and identify them transcriptome-wide, it captures only a few hundred cells (>300) in one experiment, which can be overcome with future modifications [82]. Moreover, combining long- and short-read sequencing in single cells and isolated nuclei has revealed new messenger RNAs, approximately three-fourths of the brain transcriptome, in neurodegenerative diseases, such as Alzheimer’s disease, dementia with Lewy bodies, or Parkinson’s disease [83].

RNA accomplishes many functions through various structural levels beyond its primary and secondary structural ones, defined by nucleotide sequence and Watson–Crick pairing-based folding, respectively [84,85,86,87]. Along with the staggering number of noncoding RNA genes, RNA structural versatility underlies biological diversity from the organismal to population levels.

RNA’s bases closely stack on each other like ‘coins in a roll’ via noncovalent interactions, exposing their charged exocyclic groups to water molecules and ions, underlying RNA’s solvability and helical conformations unrelated to Watson–Crick pairing [88]. RNA’s conformation also varies with environmental changes, liquid-liquid phase separation, or interactions with other biomolecules [89,90,91].

Underlying its compactness, RNA intrinsically tends to form A-U, G-C, and G-U Watson–Crick base pairs in short and long-range structures, higher-order architectures, and RNA-RNA interactions in picoseconds to seconds [92,93,94,95], which are fundamental to its diverse functions [96]. As many as 40% of the nucleotides of an RNA molecule can be part of hairpins and multi-helix junction loops [97], and 30% to 40% of RNA duplexes in living cells involve sequences over 200 nucleotides apart [96].

## 3. RNA’s Structure Is Defined at Primary, Secondary, Tertiary, Quaternary, and Quinary Levels

### 3.1. Primary, Secondary, and Tertiary RNA Structures

As illustrated in Figure 2 for the *Escherichia coli* transfer RNA for phenylalanine (tRNA^Phe^) [98], the primary structure is the RNA’s linear nucleotide sequence. The secondary structure describes the paired and unpaired elements of stems, loops, and bulges that form as the single-stranded RNA molecule folds back on itself via Watson–Crick pairs and interacts via hydrogen bonding and stacking as soon as it is synthesized [90,94] (Figure 2).

The tertiary or 3D structure, which typically compacts the RNA, is achieved by longer-distance Watson–Crick and non-Watson–Crick interactions of elements within the preformed secondary structures [86] (Figure 2).

Many RNA loops are characterized by structural modules with highly organized networks of noncanonical interactions comprising ordered non-Watson–Crick base pairs embedded between Watson–Crick base pairs [99]. Non-Watson–Crick pairs are key for folding and binding to proteins or other ligands [100,101,102].

RNA three-dimensional (3D) motifs occupy places in structured RNA molecules corresponding to the hairpin, internal, and multi-helix junction loops of their 2D structure representations [97]. These 3D structural RNA modules, with specific loop geometries, contribute to structural stability, have central roles as architectural organizers of catalytic activity and ligand binding sites in RNA molecules, and are recurrently observed in RNA families throughout phylogeny [97,103,104,105,106,107,108].

Among RNA 3D motifs are pseudoknots, which are minimally composed of two helical segments connected by single-stranded regions or loops. These interactions lock together two stem-loops by base pairing and sugar-phosphate interactions, often in a so-called kissing interaction (Figure 3A). Pseudoknots form the catalytic core of various ribozymes, self-splicing introns, and telomerase, and alter gene expression by inducing ribosomal frameshifting in many viruses (reviewed in [109]). The best characterized is the H-type pseudoknot (Figure 3A).

Ribozymes are RNA enzymes that catalyze essential cellular reactions and are potential therapeutic agents. Nucleolytic ribozymes catalyze phosphoryl transfer reactions. The nucleolytic hammerhead ribozyme containing a pseudoknot, shown in Figure 3B, is a widespread example, including within the human genome [110]. Catalytic RNA molecules possess simultaneously a genotype and a phenotype, bypassing protein expression as the determinant of phenotype as per the central molecular biology dogma. Thanks to differential folding and catalytic activity, a single RNA genotype has the potential to adopt two or perhaps more distinct phenotypes [111].

As another example of 3D motifs, guanine-rich regions in RNA and DNA can form noncanonical G-quadruplex structures encompassing stacked guanine tetrads, a square planar structure formed by four guanine residues [112] (Figure 3C). RNA G-quadruplexes participate in translation, splicing, RNA stability, and cellular stress responses, among other functions mediated by the RNA binding proteins with which they interact [112]. In synucleinopathies, including Parkinson’s disease, dementia with Lewy bodies, and multiple system atrophy, triggered by α-synuclein aggregation leading to progressive neurodegeneration, calcium influx-induced RNA G-quadruplex assembly accelerates α-synuclein phase transition and aggregation, rendering it a therapeutic target [113]. To this end, 5-Aminolevulinic acid inhibits the liquid-liquid phase separation of RNA G-quadruplexes, thereby reducing α-synuclein aggregation and associated neurodegeneration [113].

### 3.2. Quaternary RNA Structure

Similar to the interactions of DNA and histones, RNA’s quaternary structures result from a folded RNA’s interaction with other biopolymers, such as proteins and RNAs. Figure 4 shows the cyclodipeptide synthase from *Candidatus Glomeribacter gigasporarum* bound to the *Escherichia coli* phenylalanyl-tRNA^Phe^ [98]. Cyclopeptidases hijack amino acyl-tRNAs from canonical ribosomal protein synthesis to catalyze the synthesis of various cyclodipeptides [98].

The diverse tertiary structures of transfer RNA deciphered in 1974, the hammerhead ribozyme in 1993, the P4-P6 domain of the group I intron in 1996, the Hepatitis Delta Virus ribozyme in 1998, and the hairpin ribozyme in 2001 do not oligomerize into symmetric quaternary structures, as do many proteins [114,115,116,117,118]. Even the ribosome, an RNA heterotrimer comprising 5S, 16S, and 23S rRNAs, in addition to numerous ribosomal proteins, exhibits no point-group symmetry [119,120,121,122] except for the peptidyl transferase center, which displays local pseudosymmetry [123]. However, several biological RNAs, such as the bacteriophage ϕ29 prohead RNA, exhibit global symmetry at the tertiary and quaternary structural levels. Global symmetry stabilizes the RNA fold, coordinates ligand-RNA interactions, and facilitates association with symmetric binding partners [124].

### 3.3. Quinary RNA Structure

The quinary structure of RNA results from its weak and nonspecific interaction with cellular metabolites, such as osmolytes, accumulated by cells in response to osmotic stress [91]. Understanding the effects of osmolytes on RNA’s tertiary structure, whether stabilizing or destabilizing, is crucial to comprehend the intricacies of RNA [125]. For instance, as hydrated magnesium ions neutralize a notable fraction of the negative charge of an RNA tertiary structure, the RNA becomes less responsive to stabilizing osmolytes and may even be destabilized [125].

Real-time in-cell nuclear magnetic spectroscopy reveals that RNA can also modulate protein quinary structure. This is exemplified by quinary interactions between the thioredoxin protein and messenger RNA fostered by antibiotics, such as tetracycline and streptomycin, that bind the bacterial small ribosomal (30S) subunit [126]. Messenger RNA and ribosomes, representing up to 90% of the total RNA in the cell, prominently mediate these weak, cytosolic quinary protein interactions, which affect protein stability, substrate binding, and activity [127,128,129,130,131,132,133,134,135,136,137]. As another example, the enzymes adenylate kinase and dihydrofolate reductase, and the respective coenzymes, ATP and NADPH, bind to ribosomes with micromolar affinity, suppressing both enzymes’ activities [135].

### 3.4. Determining RNA Tertiary and Beyond Structures Remains Challenging

The complex biological functions of RNA molecules are underpinned by their specific sustained 3D structures, with or without the help of proteins or other RNAs in multimolecular complexes [138]. However, the study of RNA 3D structure is often hindered by the scarcity of atomic coordinates, a significant challenge in the field. These determinations are typically low-resolution or miss atoms due to the limitations of the low-throughput and costly structure determination methods, i.e., X-ray crystallography, nuclear magnetic resonance, and cryo-electron microscopy [139], which also creates a significant gap between the number of RNAs sequenced and the number of structures defined. Moreover, RNA’s shifting into diverse forms according to environmental conditions renders structural studies challenging. Traditional imaging methods, such as cryo-electron microscopy single-particle averaging analysis, rely on averaging data from thousands of selected molecules with common shapes, making it difficult to capture the unique shapes of individual RNA molecules.

Developed during the last two decades [140], some RNA 3D structure prediction computational tools use high-resolution homologs’ more precise structural information to annotate the base-pairing interactions in low-resolution structures in coarse-grained models/simulations [141,142] or in imaging data missing atoms [143]. Moreover, a machine-learning approach, termed Atomic Rotationally Equivariant Scorer (ARES), identifies accurate structural models without assumptions about their defining characteristics despite being trained with the atomic coordinates of only 18 known RNA structures [144]. ARES predicted RNA three-dimensional structures with accuracy surpassing both human expertise and previously established methods.

Structural imaging studies are complemented by gel or capillary electrophoresis based on in-line probing, i.e., structural sensitivity to spontaneous degradation, nucleases targeting either single- or double-stranded regions, or chemical probes, such as dimethyl sulfoxide (DMS). DMS, for instance, is used to probe unpaired adenines and cytidines, and 1-metho-p-toluenesulfonate (CMCT) to probe unpaired uridines in chemical inference of RNA structures sequencing (CRIS-seq) [145,146,147]. DMS is also used in RNA structure sequencing (Structure-seq and STRucture-seq2) [148,149,150,151], dimethyl sulfate-modified RNA sequencing (DMS-seq) [152], dimethyl sulfate mutational profiling with sequencing (DMS-MaPseq) [153], and transfer RNA structure sequencing (tRNA structure-seq) [154]. Pyrdiostatin, the chemical probe in RNA GQ sequencing (rG4-seq) [153], and selective 2′-hydroxyl acylation analyzed by primer extension (SHAPE) [155,156,157,158,159,160,161] have also extended chemical probing to the entire transcriptome [86,162,163,164,165].

Even if RNA structures are accurately determined, they may not represent the one(s) relevant in vivo. Many factors influence RNA structure in the living cell, including variations in organelle environments and interactions with proteins or other macromolecules, which render the elucidation of RNA structure in vivo particularly challenging. For instance, in silico modeling provides the most thermodynamically stable structure of an RNA sequence, while RNAs can become trapped in vivo in alternative structures [86,166]. Moreover, processing the low abundance, long nascent, or precursor RNAs, including splicing and polyadenylation, entails pathway networks that determine mature isoform composition and control gene expression, further adding to the complexity of studying RNA structure [167].

Alphafold’s success [168,169] in predicting protein 3D structures has not yet extended to RNA [170]. This is due to differences in building blocks (amino acids vs. nucleotides), diversity of sequence range (up to tens of thousands of nucleotides for RNA vs. a few hundred amino acids for proteins), number of available structure data (orders of magnitude greater for proteins), and folding stability (multiple conformations for RNA vs. usually one for proteins) [140].

Readily available RNA 3D structural prediction tools often rely on the primary sequence and canonical 2D structures formed by A-U, G-C, and G-U Watson–Crick pairs to detect structural RNA modules from primary sequence data and identify recurrent interaction networks [171,172,173,174,175,176]. Several databases contain RNA structural information [108,177,178,179].

During the last decade, computational RNA structure predictions have evolved from the earliest thermodynamic and molecular dynamic-based approaches to deep learning-based conformation approaches [180]. Earlier deep learning models for RNA structure have been competitive but not consistently better than traditional 3D structure prediction methods, including ab initio physics-based methods using various levels of granularity in nucleotide representation, template-based methods that try to map sequences to structural motifs before merging them into a whole structure, or hybrid methods, combining ab initio and template-based methods [140,181]. However, platforms such as the RNA3DB dataset [182], which arranges the RNA 3D chains into distinct non-redundant groups (Components), and Dfold, which combines an autoregressive Deep Generative Model, Monte Carlo Tree Search, and a scoring model [183] have been developed to improve RNA 3D structure prediction.

An innovative technique to study the 3D structure of individual molecules without averaging builds on advanced Individual-Particle cryo-Electron Tomography (IPET) to focus on single-molecule 3D imaging in cryopreserved samples. IPET captures a snapshot of RNA’s folding landscape by capturing molecules in various folding stages, from immature states to their optimal shape. This approach may allow the folding engineering of more effective RNA vaccines, dynamic sensors for molecular medicine [184], and advances in RNA-based gene therapy.

## 4. RNA in Gene Therapy: An Example of Structure-Function Knowledge Application

RNA-based therapeutics have emerged as a powerful subset of gene therapy, offering significant advantages in targeting previously considered “undruggable” pathways and providing versatile therapeutic options. RNA therapeutics have shown promise in treating multiple conditions, including genetic disorders, cancers, and infectious diseases [185]. In regenerative medicine, messenger RNA-based approaches have demonstrated potential for cell reprogramming and targeted tissue restoration [186].

The evolution of RNA in gene therapy has been marked by significant structural-functional discoveries and technological advances that have enhanced our understanding and therapeutic potential [187]. Key milestones include the development of antisense oligonucleotides (ASOs) in the early 1980s and the proposal of RNA interference (RNAi) in the 2000s [187,188].

### 4.1. Types of RNA Used in Gene Therapy

#### 4.1.1. Messenger RNA

mRNA is a powerful tool in gene therapy, particularly for protein replacement and vaccination strategies [189]. This approach directly delivers synthetic mRNA-encoding therapeutic proteins into cells [190]. Once there, cellular machinery translates it into functional proteins [189]. The rapid development and success of mRNA vaccines against SARS-CoV-2 have showcased the potential of this technology, accelerating research into mRNA therapeutics for other diseases [191].

The therapeutic use of mRNA offers several advantages, including transient expression without genomic integration, which enhances safety [189]; the ability to produce almost any functional protein or peptide in the human body [190,191]; faster design and production compared to conventional approaches [190]; cost-effectiveness and flexibility [190]; and higher transfection efficiency and lower toxicity compared to DNA-based drugs [190,191].

Transient expression without genomic integration enhances safety in mRNA-based therapies. Unlike DNA-based approaches, mRNA does not need to enter the nucleus to function, eliminating the risk of insertional mutagenesis [192]. This transient nature ensures that the mRNA is only active for a limited time, reducing the potential burden on host homeostasis and decreasing off-target effects in applications such as gene editing [193].

One of the main advantages of mRNA therapeutics is their ability to produce nearly any functional protein or peptide in the human body. This versatility paves the way for treatments targeting various diseases, including those previously deemed unmanageable or genetic [191,194,195]. The programmable aspect of mRNA allows for the rapid production of various proteins, making it a powerful tool for precision medicine [191].

mRNA therapeutics offer faster design and production times than traditional methods [195]. mRNA’s simplicity makes it well-suited for studying short-term genetic effects and quickly creating recombinant proteins. This advantage is significant in responding to large-scale outbreaks of infectious diseases, as evidenced by the rapid development of mRNA vaccines for COVID-19 [196,197].

Cost-effectiveness and flexibility are key factors that enhance the attractiveness of mRNA therapeutics [190]. Generally, the production and manufacturing of mRNA are less expensive and more convenient than proteins, especially when creating vaccine products during a pandemic [198,199].

mRNA therapeutics also provide greater transfection efficiency and lower toxicity when compared to DNA-based drugs [200]. Research shows that mRNA electroporation can achieve up to 98% transfection efficiency across various cell lines, significantly outperforming DNA-based systems [201]. Furthermore, mRNA transfection generally results in higher cell viability than DNA transfection, making it a safer option for cellular manipulation [200].

The history of mRNA therapeutics dates back to the early 1960s when mRNA was discovered as a critical player in genetic information flow [189,190]. However, it was in the late 1980s that researchers began exploring mRNA as a therapeutic tool [190]. In 1987, Robert Malone from the Salk Institute demonstrated that synthetic mRNA strands mixed with lipid particles could transfect human cells to express proteins of interest [190].

As the field continues to evolve, over 54 mRNA vaccines and drugs are currently in various stages of clinical testing for multiple diseases, from infectious to cardiovascular conditions [190]. The versatility and rapid production capabilities of mRNA therapeutics position them as a promising tool in the future of precision medicine.

#### 4.1.2. siRNA

Small interfering RNA (siRNA) operates through the RNA interference (RNAi) pathway to silence specific genes [188,201,202,203]. This mechanism involves double-stranded RNA molecules that bind to complementary mRNA sequences, inducing their degradation and allowing for targeted gene knockdown [188,201,202]. siRNA therapeutics offer several advantages, including targeting almost any gene with high precision [204], and silencing genes previously considered “undruggable” [188].

Andrew Fire and Craig Mello discovered RNAi in 1998, changing our understanding of gene regulation and opening new avenues for therapeutic intervention [188,202,204]. As research progresses, innovations in chemical modifications and delivery systems continue to improve the efficacy and safety of siRNA therapeutics, making them a promising tool in treating various diseases [203].

#### 4.1.3. miRNA

MicroRNA (miRNA) therapeutics have emerged as a promising approach in gene therapy [205]. They offer unique advantages in modulating gene expression patterns associated with various diseases [206]. Two main strategies have been developed to manipulate miRNA activity: miRNA mimics and inhibitors.

miRNA mimics are synthetic double-stranded RNA molecules designed to replicate the function of endogenous miRNAs [193]. They supplement downregulated or non-functional miRNAs, recruit the RNA-induced silencing complex (RISC) to complementary mRNA sequences, and facilitate targeted RNA interference [207]. Advantages of miRNA mimics include the potential to restore tumor suppressor miRNA function in cancer therapy [14], and the versatility in targeting multiple genes simultaneously [207].

Conversely, miRNA inhibitors, also known as anti-miRNAs or antagomiRs, are chemically modified, single-stranded oligonucleotides designed to bind to and inhibit endogenous miRNAs [207]. This mechanism upregulates target mRNA translation and alleviates the effects caused by the overexpression of malignant miRNAs [208]. Advantages of miRNA inhibitors include the ability to target specific miRNAs [207], and the potential for long-lasting effects due to chemical modifications [206].

miRNAs have shown promise in therapeutic applications, particularly in cancer treatment and other disorders characterized by dysregulated gene expression [205,207]. Their endogenous nature and ability to regulate multiple genes within a pathway offer potential advantages over traditional single-target therapies [207].

Several miRNAs, including miR-20a, let-7a, miR-17, miR-18a, miR-27a, and miR-92a, give rise to cleavage-inducing tiny RNAs (cityRNAs) when truncated to 14 nucleotides. As exceptionally short-guide RNAs, cityRNAs uniquely activate Argonaute proteins, particularly Argonaute 3 (AGO3) while inhibiting AGO2 for target RNA cleavage. This finding has significant implications for understanding RNA interference mechanisms and gene regulation, underlying new possible RNA-based therapeutics and research tools [209,210].

#### 4.1.4. Non-Coding RNAs

Non-coding RNAs (ncRNAs) are pivotal in regulating gene expression and cellular processes, making them essential components in the gene therapy landscape. These molecules are not merely transcriptional byproducts but integral to various biological functions, influencing epigenetic regulation, transcriptional control, and post-transcriptional modulation of gene expression [211,212,213,214,215].

In addition to their regulatory functions, ncRNAs are central to numerous cellular processes such as RNA splicing, nuclear architecture maintenance, and signal transduction pathways [212,216]. One of the key processes in which ncRNAs are involved is RNA splicing, which contributes to the precise removal of introns and the joining of exons in pre-mRNA transcripts [217,218]. This splicing is crucial for generating mature mRNA molecules that encode functional proteins, influencing gene expression outcomes.

ncRNAs are integral to maintaining nuclear architecture, as they help organize chromatin and facilitate interactions between different genomic regions [217]. This spatial organization is essential for proper gene expression and cellular function. Furthermore, ncRNAs participate in various signal transduction pathways, mediating extracellular signals to intracellular responses [219,220,221]. By influencing these pathways, ncRNAs can modulate critical cellular processes such as metabolism, immune responses, and stress responses.

ncRNAs regulate other signaling pathways that govern essential cellular functions such as proliferation, differentiation, and apoptosis [212]. For instance, specific microRNAs (miRNAs) can target mRNAs encoding proteins involved in cell cycle regulation, thereby promoting or inhibiting cell division [222,223]. This regulatory capacity is vital for maintaining tissue homeostasis and ensuring cells respond appropriately to developmental cues and environmental stimuli.

ncRNAs play a crucial role in apoptosis by modulating the expression of pro-apoptotic and anti-apoptotic factors [224]. This determines whether a cell will undergo programmed cell death or survive under stress conditions. The delicate balance ncRNAs maintain is fundamental to an organism’s health [224].

The involvement of ncRNAs in disease mechanisms has garnered significant attention due to their implications in various pathologies, including cancer, neurological disorders, and cardiovascular diseases [225,226]. Furthermore, ncRNAs have been shown to regulate processes such as cardiac hypertrophy and vascular remodeling in cardiovascular diseases [227,228]. Understanding the roles of ncRNAs in these diseases provides insights into their underlying mechanisms. It opens new avenues for developing targeted therapies that harness the unique properties of these regulatory molecules.

##### Long Non-Coding RNAs

Long non-coding RNAs (lncRNAs) are becoming significant targets and tools in gene therapy, and their potential applications are expanding rapidly [208]. Typically longer than 200 nucleotides, lncRNAs play crucial roles in gene regulation and various cellular processes, influencing complex genetic networks [208,212]. Manipulating them through gene therapy may offer new avenues for treating complex genetic disorders and cancers.

The therapeutic targeting of lncRNAs has gained traction over the past decade as their diverse functions in gene regulation have been uncovered [208,212]. LncRNAs can modulate chromatin structure, influence transcriptional and post-transcriptional processes, and interact with proteins and other RNAs, making them integral to cellular function [208,212]. For instance, some single-stranded lncRNAs influence chromatin structure by interacting with double- or single-stranded DNA. These interactions can occur as RNA•(DNA)2 triplexes, stabilized by nucleosomes prominently via their histone H3-tail component, or as R-loops. LncRNA-induced R-loops play crucial roles in plants [229]. Triplexes follow preferred patterns of palindromic polypyrimidine or polypurine stretches [212,230,231]. An atomic force microscopy-based approach to discriminate triplexes from R-loops could provide single-molecule level insight into lncRNAs’ roles in gene regulation [232].

The advantages of targeting lncRNAs include their tissue- or cell-type-specific expression patterns [212], acting as scaffolds for protein complexes, enhancers of gene expression, or decoys that inhibit oncogenic pathways [208], and usefulness as diagnostic and prognostic biomarkers [212]. For instance, the *MEG3* lncRNA with multiple conserved pseudoknots is involved in the regulatory network of the tumor suppressor p53 [233]. Some lncRNAs function as RNA hubs, as illustrated by the long form of NEAT1 [234], which appears to lack significant self-structure but may participate in long-range interaction between multiple lncRNAs [235]. Likewise, the lncRNA MALAT1 localizes to nuclear speckles and has many long-range interactions [96], including with other RNAs such as NEAT1 and U1 snRNA [236].

Ongoing research is exploring various strategies for effectively targeting lncRNAs. These strategies include transcriptional inhibition, post-transcriptional modulation, and using CRISPR technology to edit lncRNA expression patterns or genomic loci [208,212].

LncRNAs are involved in numerous diseases. For instance, studies have shown that targeting specific oncogenic lncRNAs can inhibit tumor growth and metastasis in preclinical models [194]. Furthermore, restoring downregulated or lost lncRNAs presents an exciting opportunity for therapeutic development [208].

### 4.2. Molecular and Cellular Mechanisms

#### 4.2.1. Antisense Oligonucleotides (ASOs)

ASOs are short, synthetic strands of nucleic acids that bind to complementary mRNA sequences [189,191,195]. This binding can either induce degradation of the target mRNA through RNase H-mediated cleavage (gapmers) or act through steric hindrance (mixmers) via high-affinity binding to complementary targets [237]. For example, ASOs can modulate splicing, effectively altering protein production [212,238]. By targeting specific mRNAs, ASOs can reduce the expression of disease-causing proteins [204]. As such, they are valuable tools in treating genetic disorders.

ASOs are typically well tolerated, and several ASO therapeutics have received regulatory approval for clinical use in the treatment of cytomegalovirus retinitis, homozygous familial hypercholesterolemia, Duchenne muscular dystrophy, spinal muscular atrophy, hereditary transthyretin amyloidosis, and polyneuropathy [239]. Additionally, ASO manufacturing is well established and can be readily scaled up, with chemically modified gapmer and mixmer ASOs containing interspersed nucleotides linked by phosphorothioate bonds conferring increased affinity, stability, and improved pharmacokinetic/pharmacodynamic properties [240,241].

#### 4.2.2. RNA Interference (RNAi)

RNAi employs small interfering RNAs (siRNAs) and microRNAs (miRNAs) to silence specific genes [188]. siRNAs are designed to match complementary mRNA sequences, leading to their degradation by the RNA-induced silencing complex (RISC) [239]. RNAi provides a powerful means to regulate gene expression and has significant therapeutic potential in conditions such as cancer and viral infections [188,205].

#### 4.2.3. mRNA Delivery

Synthetic mRNAs can be introduced into cells to produce therapeutic proteins [189,190]. This approach has gained significant attention with the development of mRNA vaccines, particularly during the COVID-19 pandemic [191]. Synthetic mRNAs can stimulate an immune response or replace defective proteins in genetic disorders by encoding specific proteins [242]. The ability to rapidly design and produce mRNAs allows for flexible responses to emerging health threats [190,191].

#### 4.2.4. CRISPR-Cas Systems

RNA guides are crucial in CRISPR-Cas technology, enabling precise genome editing [190,204,208,243]. In this system, a guide RNA (gRNA) directs the Cas enzyme to specific DNA sequences within the genome, allowing for targeted modifications such as gene knockouts or insertions [243]. This capability represents a significant advancement in gene therapy, offering potential cures for genetic defects rather than merely treating symptoms.

### 4.3. RNA Delivery Methods

#### 4.3.1. Viral Vectors

Viral vectors, particularly adeno-associated viruses (AAVs) and lentiviruses have become essential tools for delivering RNA-based therapies in gene therapy [244]. These vectors are adept at transducing target cells and facilitating the long-term expression of therapeutic RNAs. This makes them valuable for treating various genetic disorders and diseases. Their integration into host genomes allows for sustained therapeutic effects [244]. This ability is particularly beneficial in chronic conditions requiring ongoing protein expression.

Adeno-associated viruses (AAVs) are favored for their low immunogenicity and capacity to transduce both dividing and non-dividing cells [244]. This versatility enables AAVs to be used in a wide range of tissues, including those that are difficult to target with other delivery methods [244].

AAVs have gained prominence in gene therapy due to their exceptional safety profile and versatility. AAVs have emerged as the leading in vivo delivery system for CRISPR components compared to other viral methods [245]. AAVs have several advantages. Firstly, low immunogenicity reduces the risk of adverse immune responses [246,247,248,249]. Secondly, the ability to transduce both dividing and non-dividing cells, expanding their therapeutic potential [249]. Thirdly, broad tissue tropism enables the targeting of various cell types and tissues. At least 12 AAV serotypes and over 1000 variants have been identified, each with the potential to target different cell types [249,250,251]. Fourthly, minimal pathogenicity, as AAVs are not known to cause diseases in humans [252]. While AAV is generally considered non-pathogenic, some reports have associated AAV infection with adverse reproductive outcomes [253]. Lastly, AAV have predominantly episomal transgene expression, reducing the risk of insertional mutagenesis [254]. While AAV vectors predominantly remain episomal, low-frequency integration events can occur. These integration events are typically rare but may happen at a higher rate in dividing cells or under certain conditions [255].

Recent advances in AAV technology have significantly enhanced their utility and safety for gene therapy applications [245]. Recombinant adeno-associated virus (rAAV) vectors have been designed to alleviate concerns regarding integration into the host genome [256]. By eliminating Rep genes, the integration of rAAV becomes less efficient. This alteration significantly reduces their ability to integrate into the host genome, enhancing safety [257]. Instead, rAAV genomes predominantly persist as episomes in the nucleus, avoiding risks associated with insertional mutagenesis [257]. These vectors can either form episomes or recombine randomly with host genomes, reducing the risk of insertional mutagenesis [256]. The diminished ability for preferential integration allows for more controlled expression of the transgene.

Clinical trials and preclinical studies have shown that rAAV vectors exhibit a relatively good safety profile with no severe adverse events linked to vector-related integration [258]. However, rare random integration and dose-dependent genotoxicity cases have been observed in animal models, emphasizing the importance of vector design and dose optimization [258,259].

Certain adeno-associated virus (AAV) serotypes have demonstrated a strong ability to target the central nervous system (CNS) [260]. Notably, AAV8 and AAV9 exhibit robust axonal transport capabilities, which enhance their effectiveness in transducing neurons. Both serotypes show anterograde and retrograde transport within nonreciprocal circuits following injection into an adult mouse’s brain, with similar distal transduction patterns [260].

These serotypes can cross the blood-brain barrier (BBB), expanding their potential for therapies to treat CNS conditions [261]. This ability allows for less invasive delivery methods when addressing neurological disorders. The effectiveness of these serotypes in crossing the BBB and targeting the CNS has been demonstrated in rodents and larger animal models, including non-human primates [261,262].

While these serotypes show great promise, their efficiency can vary based on the specific brain region, the species, and the type of cells targeted [262]. Ongoing research continues to refine and enhance these vectors for more effective targeting of the CNS.

Significant progress has been made in engineering adeno-associated virus (AAV) capsids to enhance gene transfer to the central nervous system (CNS) [263]. A minimal footprint from the AAVrh.10 capsid has been identified and incorporated into the AAV1 capsid, enabling it to cross the blood-brain barrier [263]. The engineered capsid, AAV1RX, demonstrates extensive neuronal transduction while reducing transduction in vascular and hepatic tissues [263]. This approach provides a roadmap for designing synthetic AAV capsids with improved targeting of the CNS and enhanced safety profiles.

In contrast to AAVs, lentiviruses are capable of stable integration into the host genome, which is advantageous for achieving persistent gene expression [244]. These properties make viral vectors a cornerstone in developing RNA-based gene therapies.

Lentiviruses offer unique advantages in gene therapy, particularly for applications requiring stable, long-term transgene expression. Lentiviruses can infect both dividing and non-dividing cells, including stem cells and neurons [264,265]. Lentiviruses can integrate their DNA into the host cell’s genome, allowing for long-term gene expression, especially in dividing cells [266,267]. This integration is facilitated by regulatory elements such as long terminal repeats (LTRs) found in lentivirus packaging plasmids [266,268]. This capability for stable integration ensures persistent gene expression, which is particularly beneficial for treating diseases that require continuous production of therapeutic proteins [268,269]. Moreover, lentiviral vectors have a large payload capacity, enabling the delivery of larger therapeutic genes [270]. They can package approximately 8 to 12 kilobases (kb) of foreign DNA, making them suitable for delivering more complex or larger therapeutic genes [271].

Recent advances in lentiviral vector technology include the development of non-integrating lentiviral vectors (NILVs). NILVs have been created to retain the advantages of lentiviruses while minimizing the risk of insertional mutagenesis. These vectors are produced by introducing mutations in the viral enzyme integrase and/or altering the viral DNA that integrase recognizes [272].

NILVs can either stably express transgenes from episomal DNA in non-dividing cells or do so transiently if the target cells are dividing [272]. They are particularly useful for post-mitotic tissues like the retina, brain, and muscle [272]. They have been shown to transduce multiple cell types and tissues, making them ideal vectors for vaccination and immunotherapies.

Modern lentiviral vectors have demonstrated improved safety profiles in clinical trials. Self-inactivating (SIN) lentiviral vectors have been shown to pose a lower risk of insertional mutagenesis compared to γ-retroviral vectors [273]. However, it is important to note that even SIN lentiviral vectors with therapeutically relevant enhancers/promoters in internal positions can still be genotoxic in sensitized mouse models [273].

Although the risk of insertional mutagenesis has not been completely removed, there have been no reported cases of leukemogenesis in hematopoietic stem cells or T-cell modification trials using modern lentiviral vectors [274]. This enhanced safety profile is due to advancements in vector design, such as self-inactivating long terminal repeats (SIN LTRs) and careful selection of internal promoters [274].

Despite the aforementioned improvements, ongoing research continues to refine and enhance the safety of lentiviral vectors [273]. For example, incorporating engineered chromatin insulator cassettes has effectively reduced a lentiviral vector’s ability to activate oncogenes through enhancer-mediated mechanisms [273].

The choice between these vectors depends on factors such as the target tissue, desired duration of expression, and safety considerations. Both AAVs and lentiviruses offer distinct advantages for different therapeutic applications. AAVs are preferred for in vivo gene delivery due to their excellent safety profile and ability to provide long-term expression without integration [245]. Due to their integration capabilities and larger payload capacity, lentiviruses are particularly useful for ex vivo gene therapy approaches, such as modifying hematopoietic stem cells [268].

#### 4.3.2. Non-Viral Vectors

Non-viral delivery systems have gained significant prominence in RNA-based gene therapy, mainly through lipid nanoparticles (LNPs) [190,275]. LNPs are designed to effectively encapsulate and protect RNA molecules, facilitating cellular uptake and enhancing endosomal escape [275]. This capability ensures therapeutic RNA reaches its intended target within the cell, maximizing its efficacy.

LNPs typically comprise four main components: ionizable lipids, helper or neutral lipids, cholesterol, and PEGylated lipids [276]. Ionizable lipids are crucial as they bind and encapsulate RNA, facilitating endosomal escape by changing charge based on pH [275,277]. Helper lipids, often phospholipids like DSPC, provide structural stability and enhance delivery efficacy [275,278]. Cholesterol contributes to the overall stability of LNPs and aids in membrane fusion [276,278]. At the same time, PEGylated lipids stabilize the LNPs during formulation and storage, modulating immune responses and pharmacokinetics [276].

LNPs offer several key advantages for RNA delivery. They protect RNA from degradation by encapsulating and shielding it from extracellular ribonucleases [277,279]. Their design allows for efficient cellular uptake, as the ionizable lipids enable high encapsulation efficiency and effective cell entry [280]. Additionally, LNPs enhance endosomal escape, allowing for the cytosolic release of RNA by interacting with endosomal membranes [277]. They also exhibit reduced immunogenicity compared to viral vectors, making them safer for clinical applications [281]. Furthermore, LNPs are versatile in their ability to encapsulate various types of RNA and can be engineered for specific tissue targeting [282].

The development of lipid nanoparticles has been transformative for RNA delivery. This is especially true in the context of mRNA therapeutics [275]. LNPs protect RNA from degradation and promote efficient cellular uptake [190]. The protection and promotions work to address critical challenges related to RNA stability and delivery [191]. The success of LNP-delivered mRNA vaccines against COVID-19 has further validated this approach, demonstrating the potential of non-viral vectors in achieving effective therapeutic outcomes [189,190,191].

Polymeric nanoparticles have emerged as a significant synthetic carrier for RNA delivery, using biodegradable polymers to encapsulate and protect RNA molecules [283,284]. These systems provide essential stability and protection, effectively shielding RNA from degradation while enabling efficient cellular uptake and extended circulation in the bloodstream.

Various polymeric nanoparticle platforms have been explored, including polymeric micelles, dendrimers, and polymer-drug conjugates [284]. Among these, cationic polymers have garnered particular attention due to their ability to complex with negatively charged RNA molecules, facilitating their delivery into cells [283,285].

Cationic polymers such as polyethyleneimine (PEI), poly-L-lysine (PLL), poly(β-amino esters) (PBAE), and polyamidoamine (PAMAM) dendrimers are notable examples [283]. These polyplexes, typically around a few hundred nanometers, are taken up by cells through various endocytosis mechanisms [286]. These polymers can be tailored to respond to specific stimuli, allowing for more controlled and targeted RNA delivery, which enhances therapeutic efficacy.

For instance, PEI is widely used due to its high transfection efficiency. Its strong positive charge condenses RNA into nanoparticles, promoting interaction with negatively charged cell membranes and facilitating cellular uptake [285]. However, the use of PEI is often limited by its associated cytotoxicity, which can be mitigated through modifications such as PEGylation [287].

Polymeric nanoparticles’ versatility allows them to effectively encapsulate various types of RNA [287]. They can also be engineered for improved targeting and reduced toxicity while maintaining high transfection efficiency [288,289,290]. While polymeric nanoparticles show promise, lipid nanoparticles (LNPs) are currently the most advanced and widely used delivery systems for RNA therapeutics [278,290].

In addition to polymeric nanoparticles, inorganic nanoparticles such as gold, silica, iron oxide, and carbon-based materials provide versatile platforms for RNA delivery [283,291]. These nanocarriers possess unique properties, including intrinsic magnetic and optical characteristics, structural diversity, and exceptional control over their nanostructural properties [292].

For instance, gold nanoparticles (AuNPs) have been utilized to create self-assembled capsules [292]. These deliver small interfering RNA (siRNA), effectively silencing macrophage TNF-α expression [292]. This illustrates the potential of inorganic nanoparticles to achieve targeted therapeutic outcomes. They also exhibit unique optical properties and can form structures like gold nanorods (AuNRs) for controlled payload release upon irradiation [292,293]. It is possible to enhance delivery capabilities using thiolated oligonucleotides, which allow for selective binding and improved stability [294]. Gold nanoparticles (AuNPs) effectively deliver siRNA to cells, resulting in significant gene knockdown effects [295].

Silica nanoparticles are also prominent in RNA delivery applications [283]. Their tunable surface chemistry allows for functionalization by targeting ligands or drugs, enhancing their therapeutic efficacy [296]. They offer a high surface area for cargo encapsulation and can be engineered for controlled release [297]. Mesoporous silica nanoparticles (MSNs) have been specifically highlighted for their ability to carry siRNA and downregulate genes associated with osteoporosis-related diseases [295,298].

Using N-acetylgalactosamine (GalNAc) conjugates for delivering small interfering RNA (siRNA) represents an innovative strategy in targeted RNA therapies, particularly for liver applications [299]. GalNAc can be attached to siRNA molecules to enable specific delivery to hepatocytes by binding to the asialoglycoprotein receptor (ASGPR), which is highly expressed in liver cells [299,300]. This targeted approach results in rapid endocytosis of the conjugates, enhancing their therapeutic effectiveness [300].

These conjugates can increase hepatocyte delivery by approximately tenfold compared to free siRNAs, significantly enhancing gene silencing and therapeutic outcomes [300]. The conjugation of GalNAc stabilizes the siRNA and facilitates its internalization into cells, where it can exert its RNA interference (RNAi) effects [299].

Biomimetic nanovectors have gained significant attention for their ability to mimic natural cellular structures, enhancing delivery efficiency and reducing immunogenicity [301]. These innovative nano-drug delivery systems combine the low immunogenicity of biological membranes with the flexibility of synthetic nanocarriers, improving drug delivery and minimizing adverse reactions [301]. This makes them particularly promising for precision tumor therapy.

Biomimetic nano-drug delivery systems (BNDDS) utilize bio-nanotechnology to encapsulate synthetic nanoparticles within biomimetic membranes. This integration combines the beneficial properties of biological membranes, such as low toxicity and high tumor targeting, with the adaptability of synthetic carriers [301]. These systems can overcome biological barriers and achieve precise drug delivery, significantly enhancing therapeutic outcomes in cancer treatment [302,303].

Among the various biomimetic nanoparticles, those coated with cell membranes from red blood cells or cancer cells exhibit enhanced circulation times and targeted delivery capabilities [304]. These membranes allow the nanoparticles to evade immune detection and preferentially accumulate at tumor sites, thereby improving therapeutic efficacy [302]. Additionally, nanoparticles designed to mimic specific cell types can effectively target tumors by leveraging unique markers present on cancer cell surfaces [302].

Peptide-based vectors are a promising strategy for RNA delivery. They primarily consist of cationic peptides rich in essential amino acids like lysine, arginine, and histidine. These peptides function as DNA-binding units or facilitate cellular uptake [305,306].

A notable example is PepFect 6 (PF6), a cell-penetrating peptide with pH-titratable trifluoromethyl quinoline moieties, which enhances endosomal release and leads to robust RNA interference (RNAi) responses across various cell types [307]. PF6 has been specifically engineered to improve endosomal escape, addressing a significant challenge in siRNA delivery [307]. PF6/siRNA nanoparticles can penetrate entire cell populations and facilitate efficient endosomal escape, resulting in effective RNAi responses [307].

RNA delivery has seen progress with the development of various non-viral vector systems. Each method—ranging from polymeric and inorganic nanoparticles to GalNAc conjugates, biomimetic nanovectors, and peptide-based vectors—has advantages and challenges. Ongoing research aims to enhance these systems for better efficacy, safety, and specificity in delivering RNA therapeutics. Table 1 compares features between viral and non-viral vectors for RNA delivery.

### 4.4. RNA-Based Gene Therapy Applications

#### 4.4.1. Genetic Disorders

Antisense oligonucleotides (ASOs) and small interfering RNAs (siRNAs) are critical approaches developed to silence mutant genes responsible for conditions such as hereditary transthyretin-mediated amyloidosis (hATTR) [204,205]. The approval of the first siRNA drug for hATTR in 2018 marked a milestone in RNA-based gene therapy, demonstrating the potential of these therapies to address previously challenging genetic diseases [204].

ASOs bind to complementary mRNA sequences [203]. This leads to either degradation of the target mRNA or modulation of splicing processes. It allows for correcting RNA processing errors and restoring average protein production [203]. For instance, nusinersen, an ASO approved for spinal muscular atrophy (SMA), enhances exon inclusion in the SMN2 gene to compensate for the loss of function in the SMN1 gene [308].

siRNAs operate through RNA interference (RNAi) pathways, effectively silencing specific genes by targeting their mRNA for degradation [188,203]. This approach is particularly beneficial for conditions characterized by toxic protein accumulation, such as hATTR, where siRNAs can selectively inhibit harmful gene expression [204].

In addition to ASOs and siRNAs, mRNA-based therapies are being explored for protein replacement in disorders like cystic fibrosis and ornithine transcarbamylase deficiency [191]. These therapies involve delivering synthetic mRNA that encodes functional proteins directly into cells, enabling the production of therapeutic proteins that restore normal cellular function [189]. The mutation-agnostic nature of mRNA therapies allows them to be applicable across various genetic mutations, presenting a versatile option for treating monogenic diseases [190,191].

#### 4.4.2. Cancer

RNA-based gene therapies offer innovative strategies for cancer treatment, including silencing oncogenes, restoring tumor suppressor functions, and modulating immune responses [208]. Small interfering RNAs and antisense oligonucleotides are also prominent RNA-based strategies for cancer [188,204,205]. They have shown promise in preclinical and early clinical studies [204]. SiRNAs can specifically target and degrade mRNA transcripts from oncogenic fusion proteins, effectively silencing genes that drive tumor growth [190]. Similarly, ASOs can bind to complementary mRNA sequences to restore standard gene expression patterns [309].

mRNA-based therapies are also being explored for their potential in cancer treatment [190,191]. These therapies encode functional proteins that restore normal cellular functions or stimulate immune responses against tumors [208]. For instance, mRNA vaccines can elicit robust immune responses by encoding tumor-associated antigens (TAAs), allowing simultaneous delivery of multiple antigens to enhance both humoral and cell-mediated immunity [190,191].

#### 4.4.3. Infectious Diseases

Beyond vaccines, RNA therapeutics are being explored for direct antiviral effects [310]. Small interfering RNAs can target specific viral genes, effectively inhibiting viral replication by silencing essential genes required for the virus’s life cycle [310]. siRNAs can significantly reduce viral loads in various infection models [204]. This is paving the way for therapeutic applications against viruses.

The versatility of mRNA technology allows the rapid design of constructs tailored to specific pathogens, enabling quick responses to emerging infectious threats [190]. Additionally, lipid nanoparticles have been instrumental in the success of mRNA vaccines by protecting fragile mRNA from degradation and facilitating cellular uptake [189].

#### 4.4.4. Protein Malfunction Diseases

RNA-based therapies are emerging as practical solutions for diseases caused by protein malfunction [311]. They deliver functional RNA to produce missing or defective proteins. A notable example is MRT5005, an mRNA therapy designed for cystic fibrosis (CF), which has entered clinical trials as a protein replacement therapy targeting the cystic fibrosis transmembrane conductance regulator (CFTR) protein [311].

MRT5005 is a codon-optimized mRNA therapy delivered via aerosolized lipid nanoparticles directly into the lungs [191]. This genotype-agnostic approach aims to re-store CFTR protein production, essential for maintaining proper ion transport and flu-id balance in epithelial cells. Thus, it improves lung function in CF patients [311].

The development of mRNA therapies like MRT5005 reflects a broader trend toward addressing genetic disorders directly through protein replacement strategies. Unlike traditional small molecule drugs that may only target specific mutations or pathways, mRNA therapies can theoretically encode any missing protein, offering a more comprehensive solution [190,191].

#### 4.4.5. Cell Reprogramming and Tissue Restoration

mRNA tools have demonstrated significant utility in cell reprogramming approaches within regenerative medicine [189,191]. By delivering specific mRNAs that induce crucial factors necessary for reprogramming or tissue restoration, researchers can achieve greater control and safety than traditional gene therapy techniques since mRNA is transient and does not integrate into the genome [189]. This method has emerged as a promising strategy for generating induced pluripotent stem cells (iPSCs) without leaving residual genetic material after achieving desired cellular states [191].

mRNA tools enable researchers to precisely adjust the expression levels of reprogramming factors, which enhances their use in diverse cell types and conditions within regenerative medicine [190]. Beyond generating induced pluripotent stem cells (iPSCs), these mRNA tools are also being investigated for tissue engineering applications. They deliver growth factors or signaling molecules directly to damaged tissues, stimulating regeneration and promoting healing.

### 4.5. Challenges and Limitations

#### 4.5.1. RNA Stability

A primary challenge in RNA-based gene therapy is the inherent instability of RNA molecules, which are susceptible to degradation by endogenous nucleases [190,191,312]. To address this issue, researchers have developed various chemical modifications and advanced delivery systems to enhance RNA stability and protect therapeutic RNA from degradation [313]. These improvements are crucial for maximizing the pharmacological efficacy of RNA therapies and ensuring their successful application in clinical settings.

#### 4.5.2. Off-Target Effects

Ensuring the specificity of RNA therapeutics is crucial to avoid unintended gene silencing or activation [207,208]. This can lead to adverse effects. Researchers employ improved design algorithms and chemical modifications to enhance target specificity and reduce off-target effects [204,208].

#### 4.5.3. Immune Responses

Another limitation of RNA-based therapies is the potential for immune responses against the delivered RNA molecules or their delivery vehicles [208]. The immune system may recognize foreign RNA as a threat, leading to inflammatory reactions that can diminish therapeutic efficacy or cause adverse effects [208]. Strategies to mitigate these immune responses, such as employing modified nucleotides or optimizing delivery formulations, are critical for improving the safety profile of RNA therapeutics.

#### 4.5.4. Delivery

Effective delivery of RNA therapeutics to target cells remains a significant hurdle [188,190,191,207]. Traditional methods, such as viral vectors, pose risks of immunogenicity and insertional mutagenesis [244]. Consequently, non-viral delivery systems, including lipid nanoparticles (LNPs) and polymer-based carriers, are optimized to enhance cellular uptake and stability while minimizing potential side effects [190,275]. Developing these delivery mechanisms is essential for achieving the desired therapeutic outcomes in various applications.

#### 4.5.5. Expression Duration

The transient nature of mRNA therapies can be both an advantage and a limitation [190]. While the lack of genomic integration reduces long-term risks associated with permanent alterations, it also necessitates repeated administrations to maintain therapeutic effects. Developing sustained-release formulations or alternative strategies that extend the duration of action without compromising safety is an ongoing area of research [314].

#### 4.5.6. Specificity of Action and Other Challenges

Secondary structures and high similarity within sequence families render specific knockdown of particular noncoding RNAs challenging. Target site specificity can limit the application of several modalities of RNA-based therapeutics. In infectious diseases, there is also the challenge of microbial mutational escape.

### 4.6. Future Gene Therapy Perspectives

The field of RNA therapeutics is rapidly evolving, with emerging technologies poised to enhance the potency and specificity of RNA-based therapies. Innovations such as untranslated region optimization and machine learning-based synthetic RNA motif design are expected to improve therapeutic outcomes significantly. The evolution of artificial intelligence in RNA therapy has progressed from simple predictive models to sophisticated design engines. Artificial intelligence tools are increasingly utilized in RNA therapy for various purposes, including RNA structure prediction as discussed for the Atomic Rotationally Equivariant Scorer (ARES), which has demonstrated high accuracy in predicting RNA structures, surpassing previous methods [144]. For sequence optimization, machine learning approaches are used to analyze large datasets of RNA sequences and their functional outcomes to identify optimal patterns for specific therapeutic applications [315]. In academia and industry, novel artificial intelligence-based algorithms are being developed to predict tissue-specific regulatory mechanisms of RNA expression and target identification. Among the latter are binding sites of proteins and microRNAs, chemical library screening, hit-to-lead optimization, and personalized RNA therapies tailored to patients’ genetic profiles and disease characteristics.

While artificial intelligence tools show promise in accelerating the development of RNA therapeutics, challenges remain. These include the need for high-quality and comprehensive data, especially regarding the biological activity of RNA-targeting small molecules. Noncoding RNA analysis, multi-omics, and meta-transcriptomics data must also be included. Additionally, ensuring the interpretability and transparency of artificial intelligence models used in RNA therapy development is crucial for regulatory approval and building trust among researchers and clinicians.

The future of RNA-based gene therapy may increasingly rely on combination approaches that integrate multiple RNA modalities or combine RNA therapeutics with traditional small molecule drugs and immunotherapies [207]. These strategies could provide synergistic effects, enabling more effective treatment of complex diseases.

Advances in RNA engineering and synthetic biology are expanding the toolkit for gene therapy. For instance, CRISPR-Cas systems using guide RNAs offer unprecedented precision in genome editing, opening new possibilities for treating genetic disorders [208,243]. The guide RNAs contain hairpin structures that bind to exogenously introduced Cas9 protein and direct it to specific genomic DNA loci for targeted gene editing [316]. These technologies enhance the specificity of gene manipulation and allow for targeted interventions to address the underlying causes of diseases.

In an alternative therapeutic approach based on RNA structure, the chimeric degrader concept has been extended to the RNA field as ribonuclease targeting chimeras (RIBOTACs) [317]. RIBOTACs use synthetic small molecules that recognize secondary or tertiary RNA structures or antisense oligonucleotides that recognize RNA primary sequence as a “bait” fragment or guide arm. In contrast, the effector arm recruits the endogenous ribonuclease (RNase) L, causing degradation of the target RNA without affecting the host transcriptome [317,318,319,320].

## 5. Conclusions

Since the central molecular biology dogma was formulated almost seven decades ago, mounting evidence revealed that RNA is the main determinant of biological diversity. This is driven by RNA’s abundance, modifications, and structural versatility of its coding and noncoding versions, which occasionally overlap. RNA structure-function relationships also vary among cells in an organism, determining cellular identity. Therefore, understanding RNA’s tertiary, quaternary, and quinary structures and their functional relationships remains challenging. The challenge is driven by the ability of RNA to adopt various structures from the secondary to the quinary levels and undergo diverse modifications according to circumstances. This has quelled the impact of traditional computational approaches more than their intrinsic limitations. Advances in approaches, including the use of machine learning and artificial intelligence [144], data gatherings, such as the United States National Institutes of Health’s and National Academies of Sciences’ RNA sequencing initiative including its modifications [321], and novel imaging methods enabling more detailed RNA analysis within single cells and intact tissues [322] provide an optimistic outlook for the continued development and refinement of disruptive RNA-based approaches for medical therapy, diagnosis, and prevention, and agriculture and industrial applications [321].

Beyond the medical uses of RNA-based therapies, which is the scope of the present review, agricultural applications include silencing RNAs and messenger RNA modifications for crop yield enhancement, RNA interference to introduce favorable traits, and RNA technologies to increase resistance to pests, drought, salinity, and temperature. RNA interference and other RNA-based approaches can also favor biofuel production pathways in microbes and manufacturing biodegradable plastics to substitute thermoplastics for bioremediation of the environment. In veterinary applications, RNA-based interventions could be used for faster growth of food animals, improved animal breeds, and aquaculture. However, each of these applications has its regulatory hurdles and biosafety considerations which differ from those of genetically modified organisms. Moreover, scalability and access to RNA technology, particularly in low- and middle-income countries, remain challenging for all applications [323,324,325].

As artificial intelligence tools become increasingly integrated into RNA therapy development, regulatory agencies must adapt their frameworks to evaluate the safety and efficacy of their use effectively. One primary concern is the need for robust validation processes. Integrating artificial intelligence into RNA therapy development also raises ethical considerations such as data privacy and the potential for bias in algorithm design.

The evolution of RNA-based gene therapy has been characterized by remarkable advancements that have greatly enhanced our understanding of RNA biology, novel delivery technologies, and their therapeutic applications. Over the past few years, researchers have made significant strides in unraveling the complexities of RNA and its various forms, such as mRNA, siRNA, and miRNA. This deeper knowledge has allowed scientists to develop more effective and targeted treatments.

As efforts continue to address and overcome existing challenges, such as improving delivery mechanisms, ensuring stability, and minimizing potential off-target effects, there is growing optimism about the potential of RNA-based therapies. Researchers are actively exploring innovative approaches to maximize the efficacy of these treatments, broadening their application to a wide range of diseases, including genetic disorders, cancer, and viral infections. Moreover, the rise of personalized medicine relies heavily on the advancements in RNA therapeutics. By tailoring treatments to the unique genetic makeup of individual patients, we can potentially enhance therapeutic outcomes and reduce adverse effects, ushering in a new era of healthcare.

With ongoing research and more clinical trials demonstrating positive results, RNA therapeutics are on the verge of significantly contributing to regenerative medicine. These therapies hold the potential not only to treat diseases that were once deemed incurable but also to improve the quality of life for many individuals. As we look forward to the future of RNA-based gene therapy with its advantages and challenges (Figure 5), continued investment and exploration in this field could lead to transformative solutions in medicine.

## Figures and Tables

**Figure 1 ijms-26-00110-f001:**
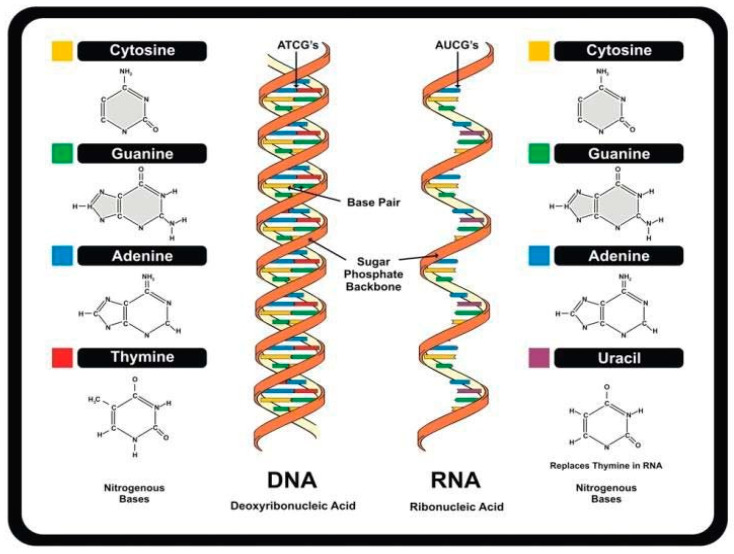
Chemical structure of nitrogenous bases, which together with the sugar and phosphate backbone, form the helical structures of nucleic acids.

**Figure 2 ijms-26-00110-f002:**
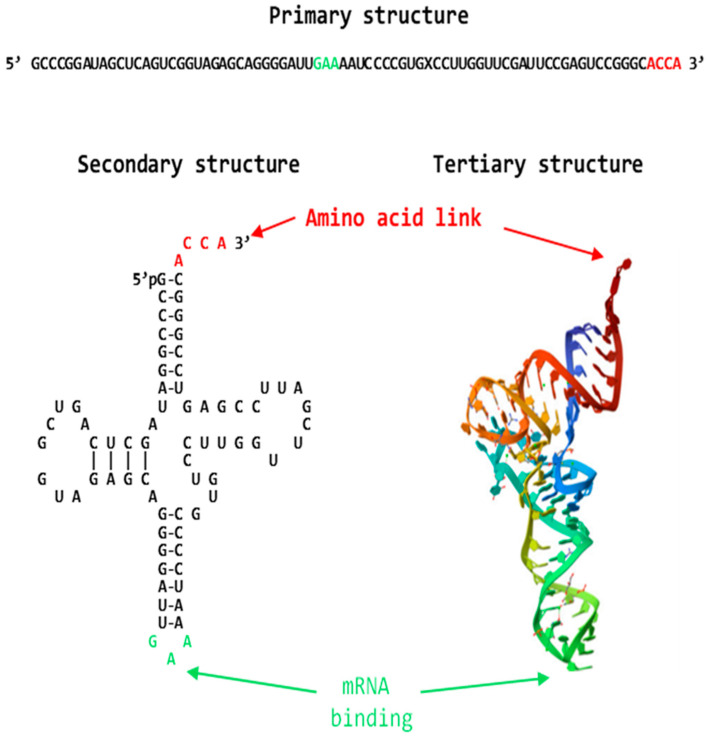
Primary, secondary, and tertiary structural levels of the *Escherichia coli* phenylalanine tRNA. The messenger RNA and amino acid binding regions are highlighted in green and red. Tertiary structure image from PDB ID:6Y3G (https://www.wwpdb.org/pdb?id=pdb_00006y3g; https://www.rcsb.org/sequence/6Y3G; both URLs accessed on 18 November 2024).

**Figure 3 ijms-26-00110-f003:**
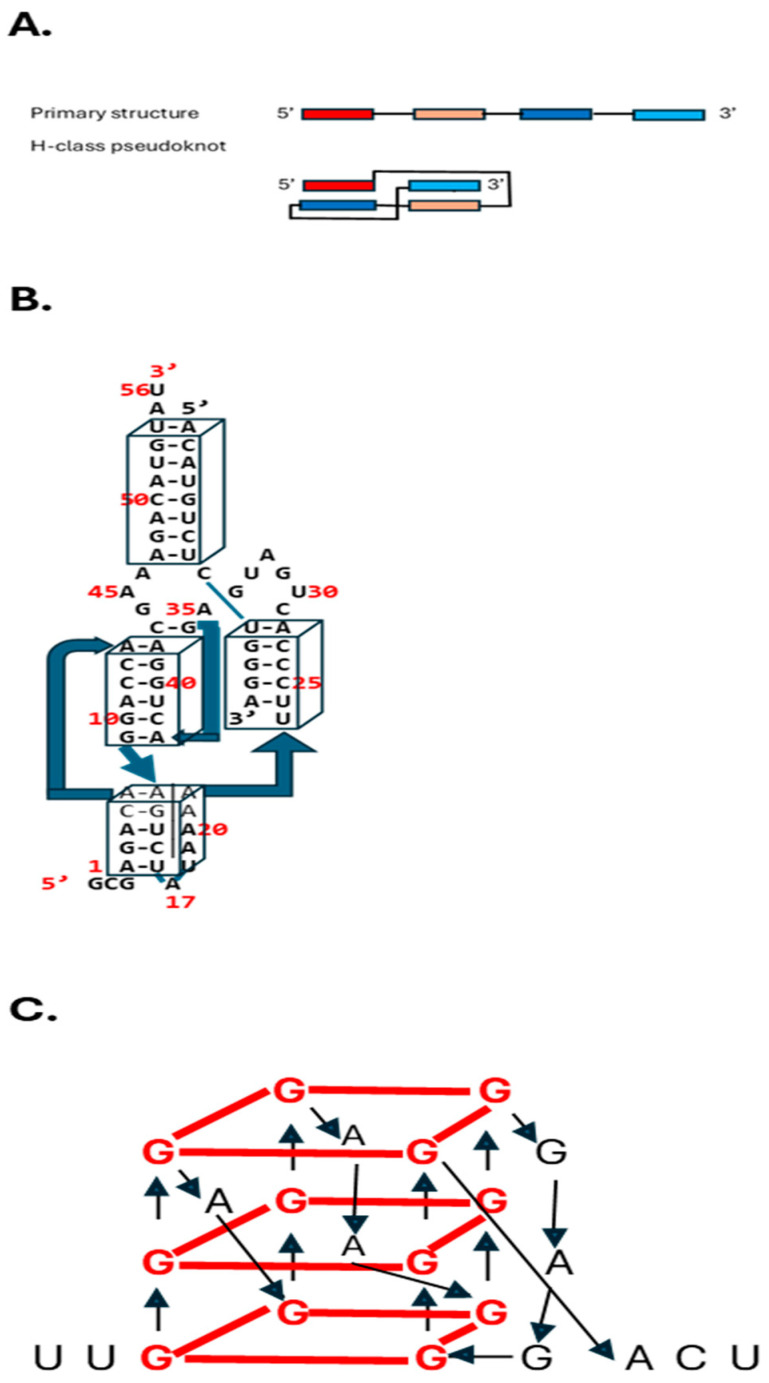
(**A**). H-type pseudoknot. Complementarity regions are shown as overlapping boxes. (**B**). Pseudoknot-containing hammerhead ribozyme with synthetic construct bound to it at the top (5′ to 3′ in black. Nucleotide positions of the ribozyme are highlighted in red, as are its 5′ and 3′ termini. (**C**). RNA G-quadruplex. Guanine residues forming stacked tetrads are in red. Arrows follow the primary sequence in panels (**B**,**C**).

**Figure 4 ijms-26-00110-f004:**
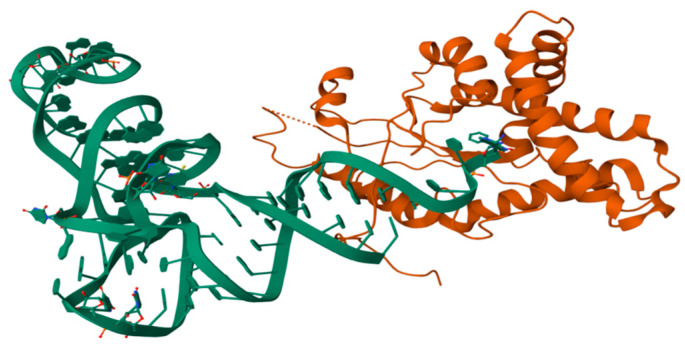
Cyclodipeptide synthase RNA-binding protein from *Candidatus Glomeribacter gigasporarum* (red) bound to the *E. coli* Phe-tRNA^Phe^ (green). Image from PDB ID:6Y4B (https://www.wwpdb.org/pdb?id=pdb_00006y4b; accessed on 18 November 2024).

**Figure 5 ijms-26-00110-f005:**
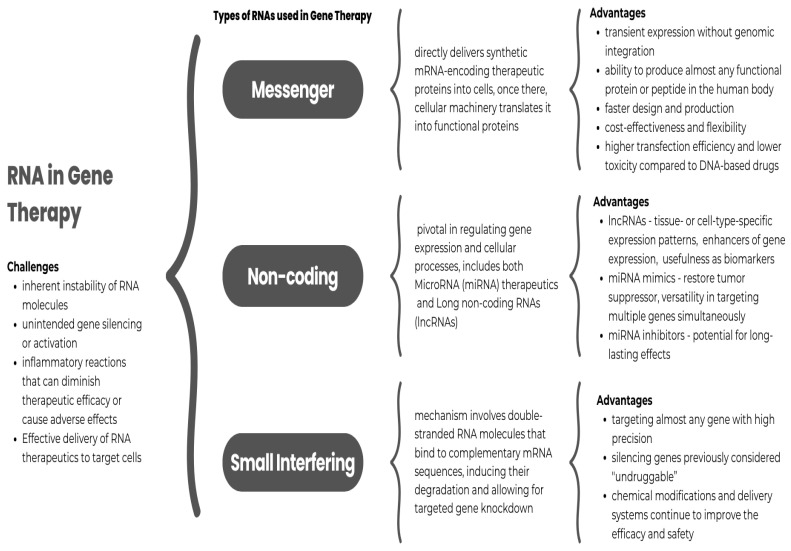
RNA in gene therapy. RNA types used, challenges, and advantages.

**Table 1 ijms-26-00110-t001:** Comparison between viral and non-viral vector-based RNA delivery methods.

Feature	Viral Vectors	Non-Viral Vectors (LNPs)
Integration	Can integrate into host genome	No integration
Immunogenicity	Low (AAVs)	Generally lower
Payload Capacity	Limited	Larger capacity
Production	Complex	Simpler
Target specificity	High (with engineered serotypes)	Can be engineered for specificity

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
