# Peer review of "RNA Structure: Past, Future, and Gene Therapy Applications"

_ijms, 2024, doi:10.3390/ijms26010110_

Round 1
Reviewer 1 Report
Comments and Suggestions for Authors
The authors have submitted a very detailed review article, which mainly summarized the RNA structure, and focusing its gene therapy applications. The RNA research history (Section 1), the various structures and challenges (Section 2, 3 and 4), the types of RNA used for therapy (Section 5.1), molecular and cellular mechanisms (Section 5.2), delivery methods and applications (Section 5.3 and 5.4), and the challenges and limitations of RNA used for therapy (Section 5.5) were summarized.
This review seems very meaningful as RNA based therapy is very promising. Surely, some issues should be carefully considered and corrected.
1. In my opinion, Section 5 is the core of this review and should be elaborated in detail, especially Section 5.3. Moreover, some more references should be cited in Section 5.
For example, Section 5.3.1, only one reference was introduced. Section 5.3.2, only LNPs was introduced.
2. Section 4 can be merged with Section 3 and its content reduced.
3. References, some references (Ref. 184, 185, 189, 192, 193, 199, 200, 204, 207) should need additional information, such as page range.
Finally, it is strongly recommended that the authors make reasonable adjustments to the content of each part, so as to introduce the structure of RNA and the application of gene therapy.
Author Response
Thank you for your time, effort, thorough review, and recommendations for improving the manuscript. Answers to your queries are below; these and other changes are highlighted in the revised manuscript.
- In my opinion, Section 5 is the core of this review and should be elaborated in detail, especially Section 5.3. Moreover, some more references should be cited in Section 5.
For example, Section 5.3.1, only one reference was introduced. Section 5.3.2, only LNPs was introduced.
We added the following sentences (highlighted in yellow) to various sections:
For microRNAs (section 4.1.3):
Several miRNAs, including miR-20a, let-7a, miR-17, miR-18a, miR-27a, and miR-92a, give rise to cleavage-inducing tiny RNAs (cityRNAs) when truncated to 14 nucleotides. As exceptionally short guide RNAs, cityRNAs uniquely activate Argonaute proteins, particularly Argonaute 3 (AGO3) while inhibiting AGO2, for target RNA cleavage. This finding has significant implications for understanding RNA interference mechanisms and gene regulation, underlying new possible RNA-based therapeutics and research tools [Park MS et al 2020, Zhang H et al 2024].
A section on non-coding RNAs was added, of which long-noncoding RNAs became a subsection (4.1.4.1)
4.1.4 Non-coding RNAs
Non-coding RNAs (ncRNAs) are pivotal in regulating gene expression and cellular processes, making them essential components in the gene therapy landscape. These molecules are not merely transcriptional byproducts but integral to various biological functions, influencing epigenetic regulation, transcriptional control, and post-transcriptional modulation of gene expression [Kaikkonen et al., 2011; Statello et al., 2021; Patil et al., 2013; Gao et al., 2020; Bu et al., 2023].
In addition to their regulatory functions, ncRNAs are central to numerous cellular processes such as RNA splicing, nuclear architecture maintenance, and signal transduction pathways [Statello et al., 2021; Chen et al., 2022]. One of the key processes in which ncRNAs are involved is RNA splicing, which contributes to the precise removal of introns and the joining of exons in pre-mRNA transcripts [Razin & Gavrilov, 2021; Frías-Lasserre & Villagra, 2017]. This splicing is crucial for generating mature mRNA molecules that encode functional proteins, influencing gene expression outcomes.
ncRNAs are integral to maintaining nuclear architecture, as they help organize chromatin and facilitate interactions between different genomic regions [Razin & Gavrilov, 2021]. This spatial organization is essential for proper gene expression and cellular function. Furthermore, ncRNAs participate in various signal transduction pathways, mediating extracellular signals to intracellular responses [Zong et al., 2023; Ramón y Cajal et al., 2019; Di et al., 2023]. By influencing these pathways, ncRNAs can modulate critical cellular processes such as metabolism, immune responses, and stress responses.
ncRNAs regulate other signaling pathways that govern essential cellular functions such as proliferation, differentiation, and apoptosis [Statello et al., 2021]. For instance, specific microRNAs (miRNAs) can target mRNAs encoding proteins involved in cell cycle regulation, thereby promoting or inhibiting cell division [Liang & He, 2011; Sati & Parhar, 2021]. This regulatory capacity is vital for maintaining tissue homeostasis and ensuring cells respond appropriately to developmental cues and environmental stimuli.
ncRNAs play a crucial role in apoptosis by modulating the expression of pro-apoptotic and anti-apoptotic factors [Ghafouri-Fard et al., 2021]. This determines whether a cell will undergo programmed cell death or survive under stress conditions. The delicate balance ncRNAs maintain is fundamental to an organism's overall health [Ghafouri-Fard et al., 2021].
The involvement of ncRNAs in disease mechanisms has garnered significant attention due to their implications in various pathologies, including cancer, neurological disorders, and cardiovascular diseases [Nemeth et al., 2023; Beňačka et al., 2023]. Furthermore, ncRNAs have been shown to regulate processes such as cardiac hypertrophy and vascular remodeling in cardiovascular diseases [Viereck et al., 2020; Kawaguchi et al., 2023]. Understanding the roles of ncRNAs in these diseases provides insights into their underlying mechanisms. It opens new avenues for developing targeted therapies that harness the unique properties of these regulatory molecules.
4.1.4.1 Long non-coding RNAs
Long non-coding RNAs (lncRNAs) are becoming significant targets and tools in gene therapy, and their potential applications are expanding rapidly [194]. Typically longer than 200 nucleotides, lncRNAs play crucial roles in gene regulation and various cellular processes, influencing complex genetic networks [194,195]. Manipulating them through gene therapy may offer new avenues for treating complex genetic disorders and cancers.
The therapeutic targeting of lncRNAs has gained traction over the past decade as their diverse functions in gene regulation have been uncovered [194,195]. LncRNAs can modulate chromatin structure, influence transcriptional and post-transcriptional processes, and interact with proteins and other RNAs, making them integral to cellular function [194,195]. For instance, some single-stranded lncRNAs influence chromatin structure by interacting with double- or single-stranded DNA. These interactions can occur as RNA•(DNA)2 triplexes, stabilized by nucleosomes prominently via their histone H3-tail component, or as R-loops. LncRNA-induced R-loops play crucial roles in plants [Gao J et al 2021]. Triplexes follow preferred patterns of palindromic polypyrimidine or polypurine stretches [Buske et al 2011, Statello et al 2021, Maldonado et al 2019, 2023]. An atomic force microscopy-based approach to discriminate triplexes from R-loops could provide single-molecule level insight into lncRNAs’ roles in gene regulation [Merici et al 2024].
The advantages of targeting lncRNAs include their tissue- or cell-type-specific expression patterns [195], acting as scaffolds for protein complexes, enhancers of gene expression, or decoys that inhibit oncogenic pathways [194], and usefulness as diagnostic and prognostic biomarkers [195]. For instance, the MEG3 lncRNA with multiple conserved pseudoknots is involved in the regulatory network of the tumor suppressor p53 [Uroda et al 2019]. Some lncRNAs function as RNA hubs, as illustrated by the long form of NEAT1 [Yamazaki et al 2018], which appears to lack significant self-structure but may participate in long-range interaction between multiple lncRNAs [Lin et al 2018]. Likewise, the lncRNA MALAT1 localizes to nuclear speckles and has many long-range interactions [Lu et al 2016], including with other RNAs such as NEAT1 and U1 snRNA [Cai et al 2020].
Ongoing research is exploring various strategies for effectively targeting lncRNAs. These strategies include transcriptional inhibition, post-transcriptional modulation, and using CRISPR technology to edit lncRNA expression patterns or genomic loci [194,195].
LncRNAs are involved in numerous diseases. For instance, studies have shown that targeting specific oncogenic lncRNAs can inhibit tumor growth and metastasis in preclinical models [194]. Furthermore, restoring downregulated or lost lncRNAs presents an exciting opportunity for therapeutic development [194].
For antisense oligonucleotides (now section 4.2.1):
ASOs are short, synthetic strands of nucleic acids that bind to complementary mRNA sequences [189,195,196]. This binding can either induce degradation of the target mRNA through RNase H-mediated cleavage (gapmers) or act through steric hindrance (mixmers) via high-affinity binding to complementary targets [Hagerdon et al 2018]. For example, ASOs can modulate splicing, effectively altering protein production [195,196]. By targeting specific mRNAs, ASOs can reduce the expression of disease-causing proteins [190]. As such, they are valuable tools in treating genetic disorders.
ASOs are typically well tolerated, and several ASO therapeutics have received regulatory approval for clinical use in the treatment of cytomegalovirus retinitis, homozygous familial hypercholesterolemia, Duchenne muscular dystrophy, spinal muscular atrophy, hereditary transthyretin amyloidosis, and polyneuropathy [Roberts et al 2020]. Additionally, ASO manufacturing is well established and can be readily scaled up, with chemically modified gapmer and mixmer ASOs containing interspersed nucleotides linked by phosphorothioate bonds conferring increased affinity, stability, and improved pharmacokinetic/pharmacodynamic properties [Eckstein 2014, Wang G et al 1999].
For viral vectors and non-viral vectors (sections 4.3.1 and 4.3.2):
AAVs have gained prominence in gene therapy due to their exceptional safety profile and versatility. AAVs have emerged as the leading in vivo delivery system for CRISPR components compared to other viral methods [Asmamaw Mengstie, 2022]. AAVs have several advantages. Firstly, low immunogenicity, reducing the risk of adverse immune responses. [Xiao et al., 1999, Wang et al., 2019, Tuisku Suoranta et al., 2022, Au et al., 2022]. Secondly, ability to transduce both dividing and non-dividing cells, expanding their therapeutic potential [Au et al., 2022]. Thirdly, broad tissue tropism enabling the targeting of various cell types and tissues. At least 12 AAV serotypes and over 1,000 variants have been identified, each with the potential to target different cell types. [ Au et al., 2022, Wang et al., 2024, Hauck & Xiao, 2003]. Fourthly, minimal pathogenicity, as AAVs are not known to cause diseases in humans [Ronzitti et al., 2020]. While AAV is generally considered non-pathogenic, some reports have associated AAV infection with adverse reproductive outcomes [Sant'Anna & Araujo, 2022]. Lastly, AAV have predominantly episomal transgene expression, reducing the risk of insertional mutagenesis [Kreppel & Hagedorn, 2022]. While AAV vectors predominantly remain episomal, low-frequency integration events can occur. These integration events are typically rare but may happen at a higher rate in dividing cells or under certain conditions [Greig et al., 2023].
Recent advances in AAV technology have significantly enhanced their utility and safety profile for gene therapy applications [Asmamaw Mengstie, 2022]. Recombinant adeno-associated virus (rAAV) vectors have been designed to alleviate concerns regarding integration into the host genome [Chandler et al., 2017]. By eliminating Rep genes, the integration of rAAV becomes less efficient. This alteration significantly reduces their ability to integrate into the host genome, enhancing safety [Penaud-Budloo et al., 2008]. Instead, rAAV genomes predominantly persist as episomes in the nucleus, avoiding risks associated with insertional mutagenesis [Penaud-Budloo et al., 2008]. These vectors can either form episomes or recombine randomly with host genomes, reducing the risk of insertional mutagenesis [Chandler et al., 2017]. The diminished ability for preferential integration allows for more controlled expression of the transgene.
Clinical trials and preclinical studies have shown that rAAV vectors exhibit a relatively good safety profile with no severe adverse events linked to vector-related integration [Sabatino et al., 2022]. However, rare random integration and dose-dependent genotoxicity cases have been observed in animal models, emphasizing the importance of vector design and dose optimization [Sabatino et al., 2022; Maurya et al., 2022].
Certain adeno-associated virus (AAV) serotypes have demonstrated a strong ability to target the central nervous system (CNS) [Castle et al., 2014]. Notably, AAV8 and AAV9 exhibit robust axonal transport capabilities, which enhance their effectiveness in transducing neurons. Both serotypes show anterograde and retrograde transport within nonreciprocal circuits following injection into an adult mouse's brain, with similar distal transduction patterns [Castle et al., 2014].
These serotypes can cross the blood-brain barrier (BBB), expanding their potential for therapies to treat CNS conditions [Song et al., 2022]. This ability allows for less invasive delivery methods when addressing neurological disorders. The effectiveness of these serotypes in crossing the BBB and targeting the CNS has been demonstrated in rodents and larger animal models, including non-human primates [Song et al., 2022; Liu et al., 2020].
While these serotypes show great promise, their efficiency can vary based on the specific brain region, the species, and the type of cells targeted [Liu et al., 2020]. Ongoing research continues to refine and enhance these vectors for more effective targeting of the CNS.
Significant progress has been made in engineering adeno-associated virus (AAV) capsids to enhance gene transfer to the central nervous system (CNS) [Albright et al., 2018]. A minimal footprint from the AAVrh.10 capsid has been identified and incorporated into the AAV1 capsid, enabling it to cross the blood-brain barrier (BBB)[Albright et al., 2018]. The engineered capsid, AAV1RX, demonstrates extensive neuronal transduction while reducing transduction in vascular and hepatic tissues[Albright et al., 2018]. This approach provides a roadmap for designing synthetic AAV capsids with improved targeting of the CNS and enhanced safety profiles.
In contrast to AAVs, lentiviruses are capable of stable integration into the host genome, which is advantageous for achieving persistent gene expression [200]. These properties make viral vectors a cornerstone in developing RNA-based gene therapies.
Lentiviruses offer unique advantages in gene therapy, particularly for applications requiring stable, long-term transgene expression. Key features include: Lentiviruses can infect both dividing and non-dividing cells, including stem cells and neurons. [Connolly, 2002; White et al., 2017]. Lentiviruses can integrate their DNA into the host cell's genome, allowing for long-term gene expression, especially in dividing cells [Ciuffi, 2008; Dong & Kantor, 2021]. This integration is facilitated by regulatory elements such as long terminal repeats (LTRs) found in lentivirus packaging plasmids [Ciuffi, 2008; Milone & O'Doherty, 2018]. This capability for stable integration ensures persistent gene expression, which is particularly beneficial for treating diseases that require continuous production of therapeutic proteins [Milone & O'Doherty, 2018; Suleman et al., 2022]. Moreover, lentiviral vectors have a large payload capacity, enabling the delivery of larger therapeutic genes [Semple-Rowland & Berry, 2014]. They can package approximately 8 to 12 kilobases (kb) of foreign DNA, making them suitable for delivering more complex or larger therapeutic genes [Counsell et al., 2017].
Recent advances in lentiviral vector technology include the development of non-integrating lentiviral vectors (NILVs). NILVs have been created to retain the advantages of lentiviruses while minimizing the risk of insertional mutagenesis. These vectors are produced by introducing mutations in the viral enzyme integrase and/or altering the viral DNA that integrase recognizes [Luis, 2020].
NILVs can either stably express transgenes from episomal DNA in non-dividing cells or do so transiently if the target cells are dividing [Luis, 2020]. They are particularly useful for post-mitotic tissues like the retina, brain, and muscle [Luis, 2020]. They have been shown to transduce multiple cell types and tissues, making them ideal vectors for vaccination and immunotherapies.
Modern lentiviral vectors have demonstrated improved safety profiles in clinical trials. Self-inactivating (SIN) lentiviral vectors have been shown to pose a lower risk of insertional mutagenesis compared to γ-retroviral vectors [Cesana et al., 2014]. However, it is important to note that even SIN lentiviral vectors with therapeutically relevant enhancers/promoters in internal positions can still be genotoxic in sensitized mouse models [Cesana et al., 2014].
Although the risk of insertional mutagenesis has not been completely removed, there have been no reported cases of leukemogenesis in hematopoietic stem cells or T-cell modification trials using modern lentiviral vectors [Schlimgen et al., 2016]. This enhanced safety profile is due to advancements in vector design, such as self-inactivating long terminal repeats (SIN LTRs) and careful selection of internal promoters [Schlimgen et al., 2016].
Despite the aforementioned improvements, ongoing research continues to refine and enhance the safety of lentiviral vectors (Cesana et al., 2014). For example, incorporating engineered chromatin insulator cassettes has effectively reduced a lentiviral vector's ability to activate oncogenes through enhancer-mediated mechanisms (Cesana et al., 2014).
The choice between these vectors depends on factors such as the target tissue, desired duration of expression, and safety considerations. Both AAVs and lentiviruses offer distinct advantages for different therapeutic applications. AAVs are preferred for in vivo gene delivery due to their excellent safety profile and ability to provide long-term expression without integration [Asmamaw Mengstie, 2022]. Due to their integration capabilities and larger payload capacity, lentiviruses are particularly useful for ex vivo gene therapy approaches, such as modifying hematopoietic stem cells [Milone & O’Doherty, 2018].
4.3.2 Non-viral vectors
Non-viral delivery systems have gained significant prominence in RNA-based gene therapy, mainly through lipid nanoparticles (LNPs) [185,201]. LNPs are designed to effectively encapsulate and protect RNA molecules, facilitating cellular uptake and enhancing endosomal escape [201]. This capability ensures therapeutic RNA reaches its intended target within the cell, maximizing its efficacy.
These nanoparticles typically comprise four main components: ionizable lipids, helper or neutral lipids, cholesterol, and PEGylated lipids [Han et al., 2023]. Ionizable lipids are crucial as they bind and encapsulate RNA, facilitating endosomal escape by changing charge based on pH [Eygeris et al., 2021; Jung et al., 2022]. Helper lipids, often phospholipids like DSPC, provide structural stability and enhance delivery efficacy [Jung et al., 2022; Ma et al., 2024]. Cholesterol contributes to the overall stability of LNPs and aids in membrane fusion [Han et al., 2023; Ma et al., 2024]. At the same time, PEGylated lipids stabilize the LNPs during formulation and storage, modulating immune responses and pharmacokinetics [Han et al., 2023].
LNPs offer several key advantages for RNA delivery. They protect RNA from degradation by encapsulating and shielding it from extracellular ribonucleases [Eygeris et al., 2021; Mukai et al., 2022]. Their design allows for efficient cellular uptake, as the ionizable lipids enable high encapsulation efficiency and effective cell entry [Swetha et al., 2023]. Additionally, LNPs enhance endosomal escape, allowing for the cytosolic release of RNA by interacting with endosomal membranes [Eygeris et al., 2021]. They also exhibit reduced immunogenicity compared to viral vectors, making them safer for clinical applications [Ryuichi Mashima & Takada, 2022]. Furthermore, LNPs are versatile in their ability to encapsulate various types of RNA and can be engineered for specific tissue targeting [Schober et al., 2024].
The development of lipid nanoparticles has been transformative for RNA delivery. This is especially true in the context of mRNA therapeutics [201]. LNPs protect RNA from degradation and promote efficient cellular uptake [185]. The protection and promotions work to address critical challenges related to RNA stability and delivery [186]. The success of LNP-delivered mRNA vaccines against COVID-19 has further validated this approach, demonstrating the potential of non-viral vectors in achieving effective therapeutic outcomes [184-186].
Polymeric nanoparticles have emerged as a significant synthetic carrier for RNA delivery, using biodegradable polymers to encapsulate and protect RNA molecules [Mendes et al., 2022; Sristi et al., 2024]. These systems provide essential stability and protection, effectively shielding RNA from degradation while enabling efficient cellular uptake and extended circulation in the bloodstream.
Various polymeric nanoparticle platforms have been explored, including polymeric micelles, dendrimers, and polymer-drug conjugates [Sristi et al., 2024]. Among these, cationic polymers have garnered particular attention due to their ability to complex with negatively charged RNA molecules, facilitating their delivery into cells [Mendes et al., 2022; Jiang et al., 2021].
Cationic polymers such as polyethyleneimine (PEI), poly-L-lysine (PLL), poly(β-amino esters) (PBAE), and polyamidoamine (PAMAM) dendrimers are notable examples [Mendes et al., 2022]. These polyplexes, typically around a few hundred nanometers, are taken up by cells through various endocytosis mechanisms [Chen et al., 2020]. These polymers can be tailored to respond to specific stimuli, allowing for more controlled and targeted RNA delivery, which enhances therapeutic efficacy.
For instance, PEI is widely used due to its high transfection efficiency. Its strong positive charge condenses RNA into nanoparticles, promoting interaction with negatively charged cell membranes and facilitating cellular uptake [Jiang et al., 2021]. However, the use of PEI is often limited by its associated cytotoxicity, which can be mitigated through modifications such as PEGylation [Yousefi Adlsadabad et al., 2024].
Polymeric nanoparticles' versatility allows them to effectively encapsulate various types of RNA [Yousefi Adlsadabad et al., 2024]. They can also be engineered for improved targeting and reduced toxicity while maintaining high transfection efficiency [Cox et al., 2021; Devulapally & Paulmurugan, 2013]. While polymeric nanoparticles show promise, lipid nanoparticles (LNPs) are currently the most advanced and widely used delivery systems for RNA therapeutics [Mendes et al., 2022; Chatterjee et al., 2024].
In addition to polymeric nanoparticles, inorganic nanoparticles such as gold, silica, iron oxide, and carbon-based materials provide versatile platforms for RNA delivery [Varshosaz & Taymouri, 2015; Mendes et al., 2022]. These nanocarriers possess unique properties, including intrinsic magnetic and optical characteristics, structural diversity, and exceptional control over their nanostructural properties [Luther et al., 2020].
For instance, gold nanoparticles (AuNPs) have been utilized to create self-assembled capsules [Luther et al., 2020]. These deliver small interfering RNA (siRNA), effectively silencing macrophage TNF-α expression [Luther et al., 2020]. This illustrates the potential of inorganic nanoparticles to achieve targeted therapeutic outcomes. They also exhibit unique optical properties and can form structures like gold nanorods (AuNRs) for controlled payload release upon irradiation [Luther et al., 2020; Dalal Mohamed Alshangiti et al., 2023]. It is possible to enhance delivery capabilities using thiolated oligonucleotides, which allow for selective binding and improved stability [Dougan et al., 2007]. Gold nanoparticles (AuNPs) effectively deliver siRNA to cells, resulting in significant gene knockdown effects [Elizarova et al., 2023].
Silica nanoparticles are also prominent in RNA delivery applications [Mendes et al., 2022]. Their tunable surface chemistry allows for functionalization by targeting ligands or drugs, enhancing their therapeutic efficacy [Khaliq et al., 2023]. They offer a high surface area for cargo encapsulation and can be engineered for controlled release [Kazemzadeh et al., 2022]. Mesoporous silica nanoparticles (MSNs) have been specifically highlighted for their ability to carry siRNA and downregulate genes associated with osteoporosis-related diseases [Cai et al., 2023; Kazemzadeh et al., 2022].
Using N-acetylgalactosamine (GalNAc) conjugates for delivering small interfering RNA (siRNA) represents an innovative strategy in targeted RNA therapies, particularly for liver applications [Springer & Dowdy, 2018]. GalNAc can be attached to siRNA molecules to enable specific delivery to hepatocytes by binding to the asialoglycoprotein receptor (ASGPR), which is highly expressed in liver cells [Springer & Dowdy, 2018; Zhang et al., 2022]. This targeted approach results in rapid endocytosis of the conjugates, enhancing their therapeutic effectiveness [Zhang et al., 2022].
These conjugates can increase hepatocyte delivery by approximately tenfold compared to free siRNAs, significantly enhancing gene silencing and therapeutic outcomes [Zhang et al., 2022]. The conjugation of GalNAc stabilizes the siRNA and facilitates its internalization into cells, where it can exert its RNA interference (RNAi) effects [Springer & Dowdy, 2018].
Biomimetic nanovectors have gained significant attention for their ability to mimic natural cellular structures, enhancing delivery efficiency and reducing immunogenicity [Han et al., 2024]. These innovative nano-drug delivery systems combine the low immunogenicity of biological membranes with the flexibility of synthetic nanocarriers, improving drug delivery and minimizing adverse reactions [Han et al., 2024]. This makes them particularly promising for precision tumor therapy.
Biomimetic nano-drug delivery systems (BNDDS) utilize bio-nanotechnology to encapsulate synthetic nanoparticles within biomimetic membranes. This integration combines the beneficial properties of biological membranes, such as low toxicity and high tumor targeting, with the adaptability of synthetic carriers [Han et al., 2024]. These systems can overcome biological barriers and achieve precise drug delivery, significantly enhancing therapeutic outcomes in cancer treatment [Beh et al., 2021; Dehaini et al., 2016].
Among the various biomimetic nanoparticles, those coated with cell membranes from red blood cells or cancer cells exhibit enhanced circulation times and targeted delivery capabilities [Li et al., 2019]. These membranes allow the nanoparticles to evade immune detection and preferentially accumulate at tumor sites, thereby improving therapeutic efficacy [Beh et al., 2021]. Additionally, nanoparticles designed to mimic specific cell types can effectively target tumors by leveraging unique markers present on cancer cell surfaces [Beh et al., 2021].
Peptide-based vectors are a promising strategy for RNA delivery. They primarily consist of cationic peptides rich in essential amino acids like lysine, arginine, and histidine. These peptides function as DNA-binding units or facilitate cellular uptake [Yang & Luo, 2023; Beloor et al., 2015].
A notable example is PepFect 6 (PF6), a cell-penetrating peptide with pH-titratable trifluoromethyl quinoline moieties, which enhances endosomal release and leads to robust RNA interference (RNAi) responses across various cell types [Samir El Andaloussi et al., 2011]. PF6 has been specifically engineered to improve endosomal escape, addressing a significant challenge in siRNA delivery [Samir El Andaloussi et al., 2011]. PF6/siRNA nanoparticles can penetrate entire cell populations and facilitate efficient endosomal escape, resulting in effective RNAi responses [Samir El Andaloussi et al., 2011].
RNA delivery has seen progress with the development of various non-viral vector systems. Each method—ranging from polymeric and inorganic nanoparticles to GalNAc conjugates, biomimetic nanovectors, and peptide-based vectors—brings advantages and challenges. Ongoing research aims to enhance these systems for better efficacy, safety, and specificity in delivering RNA therapeutics.
Figure 4. Comparison of viral and non-viral vector-based RNA delivery methods
- Section 4 can be merged with Section 3 and its content reduced.
Sections 4 and 3 have been merged. Although some sentences were shortened, another reviewer requested more detailed comparisons of current computational tools and their limitations for RNA structure determination; an expansion of RNA structural studies in vivo, and an analysis of the specific contributions or limitations of machine learning and AI in RNA structure prediction or therapy design. We have tried to include those materials in other sections to address your and the other reviewer’s recommendations as best as possible.
We also consolidated Figures 3 and 4 into new Figure 3, which now contains an added example of 3D motifs. Another reviewer had asked for additional detail on these motifs.
- References, some references (Ref. 184, 185, 189, 192, 193, 199, 200, 204, 207) should need additional information, such as page range.
Mentioned references have been corrected. Other references were also formatted appropriately, i.e., replaced “et al.” for the full author list and other changes (refs. 183, 186, 190, and 206). Journal names are all now in bold, and journal volumes in italics.
Finally, it is strongly recommended that the authors make reasonable adjustments to the content of each part, so as to introduce the structure of RNA and the application of gene therapy.
As recommended, pertinent sentences have been added throughout the manuscript (highlighted in yellow in the revised copy). Below are several examples:
At the beginning and end of section 1 (RNA is an organic code), respectively:
Understanding organic codes is fundamental to therapeutic development.
Therefore, R-loops and interacting proteins are therapeutic targets
At the beginning of section 2 (RNA has many functions through its intricate, ubiquitous, diverse, and dynamic structure):
This has fueled the rise to prominence of RNA-based therapeutics.
To underscore the importance of RNA G-quadruplexes to pathologies and their treatment, we added the following paragraph to section 3 (RNA's structure is defined at primary, secondary, tertiary, quaternary, and quinary levels):
In synucleinopathies, including Parkinson’s disease, dementia with Lewy bodies, and multiple system atrophy, triggered by α-synuclein aggregation leading to progressive neurodegeneration, calcium influx-induced RNA G-quadruplex assembly accelerates α-synuclein phase transition and aggregation, rendering it a therapeutic target [Matsuo et 2024]. To this end, 5-Aminolevulinic acid inhibits the liquid-liquid phase separation of RNA G-quadruplexes, thereby reducing α-synuclein aggregation and associated neurodegeneration [Matsuo et 2024].
In section 4 under long noncoding RNAs:
The therapeutic targeting of lncRNAs has gained traction over the past decade as their diverse functions in gene regulation have been uncovered [194,195]. LncRNAs can modulate chromatin structure, influence transcriptional and post-transcriptional processes, and interact with proteins and other RNAs, making them integral to cellular function [194,195]. For instance, some single-stranded lncRNAs influence chromatin structure by interacting with double- or single-stranded DNA. These interactions can occur as RNA•(DNA)2 triplexes, stabilized by nucleosomes prominently via their histone H3-tail component, or as R-loops. LncRNA-induced R-loops play crucial roles in plants [Gao J et al 2021]. Triplexes follow preferred patterns of palindromic polypyrimidine or polypurine stretches [Buske et al 2011, Statello et al 2021, Maldonado et al 2019, 2023]. An atomic force microscopy-based approach to discriminate triplexes from R-loops could provide single-molecule level insight into lncRNAs’ roles in gene regulation [Merici et al 2024].
The advantages of targeting lncRNAs include their tissue- or cell-type-specific expression patterns [195], acting as scaffolds for protein complexes, enhancers of gene expression, or decoys that inhibit oncogenic pathways [194], and usefulness as diagnostic and prognostic biomarkers [195]. For instance, the MEG3 lncRNA with multiple conserved pseudoknots is involved in the regulatory network of the tumor suppressor p53 [Uroda et al 2019]. Some lncRNAs function as RNA hubs, as illustrated by the long form of NEAT1 [Yamazaki et al 2018], which appears to lack significant self-structure but may participate in long-range interaction between multiple lncRNAs [Lin et al 2018]. Likewise, the lncRNA MALAT1 localizes to nuclear speckles and has many long-range interactions [Lu et al 2016], including with other RNAs such as NEAT1 and U1 snRNA [Cai et al 2020].
For microRNAs (section 4.1.3):
Several miRNAs, including miR-20a, let-7a, miR-17, miR-18a, miR-27a, and miR-92a, give rise to cleavage-inducing tiny RNAs (cityRNAs) when truncated to 14 nucleotides. As exceptionally short guide RNAs, cityRNAs uniquely activate Argonaute proteins, particularly Argonaute 3 (AGO3) while inhibiting AGO2, for target RNA cleavage. This finding has significant implications for understanding RNA interference mechanisms and gene regulation, underlying new possible RNA-based therapeutics and research tools [Park MS et al 2020, Zhang H et al 2024].
Reviewer 2 Report
Comments and Suggestions for Authors
The manuscript provides a compelling overview of RNA structure and its applications in gene therapy. Overall, it is well-written and accessible. However, I have a few recommendations to refine this review:
1. Although the title, “RNA Structure: Past, Future, and Gene Therapy Applications,” promises a balanced discussion, the manuscript allocates significant attention to the structural hierarchy of RNA—primary, secondary, tertiary, quaternary, and quinary levels. While this provides valuable context, the role of RNA in gene therapy primarily involves sequence complementarity, such as binding to complementary RNA strands. A more detailed exploration of how secondary through quinary structures specifically influence gene therapy applications would strengthen the relevance of this discussion.
2. The quality of Figure 1 requires improvement, as it appears blurry. Adding transcription direction indicators for the RNA secondary structure within the figure would be helpful. Additionally, the color scheme used in Figure 4 seems unconventional and detracts from its impact. Since this figure is central to the manuscript, a more vivid and scientifically accurate representation is strongly recommended.
Author Response
Thank you for your time, effort, thorough review, and recommendations for improving the manuscript. Answers to your queries are below; these and other changes are highlighted in the revised manuscript.
- Although the title, “RNA Structure: Past, Future, and Gene Therapy Applications,” promises a balanced discussion, the manuscript allocates significant attention to the structural hierarchy of RNA—primary, secondary, tertiary, quaternary, and quinary levels. While this provides valuable context, the role of RNA in gene therapy primarily involves sequence complementarity, such as binding to complementary RNA strands. A more detailed exploration of how secondary through quinary structures specifically influence gene therapy applications would strengthen the relevance of this discussion.
The first half of the Conclusions section has been modified with the addition of the highlighted sentences below:
Since the central molecular biology dogma was formulated almost seven decades ago, mounting evidence revealed that RNA is the main determinant of biological diversity. This is driven by RNA’s abundance, modifications, and structural versatility of its coding and noncoding versions, which occasionally overlap. RNA structure-function relationships also vary among cells in an organism, determining cellular identity. Therefore, understanding RNA’s tertiary, quaternary, and quinary structures and their functional relationships remains challenging. The challenge is driven by the ability of RNA to adopt various structures from the secondary to the quinary level and undergo diverse modifications according to circumstances. This has quelled the impact of traditional computational approaches more than their intrinsic limitations. Advances in approaches, including the use of machine learning and artificial intelligence [139], data gatherings, such as the United States National Institutes of Health’s and National Academies of Sciences’ RNA sequencing initiative including its modifications [209], and novel imaging methods enabling more detailed RNA analysis within single cells and intact tissues [210] provide an optimistic outlook for the continued development and refinement of disruptive RNA-based approaches for medical therapy, diagnosis, and prevention, and agriculture and industrial applications [209].
Beyond the medical uses of RNA-based therapies, which is the scope of the present review, agricultural applications include silencing RNAs and messenger RNA modifications for crop yield enhancement, RNA interference to introduce favorable traits, and RNA technologies to increase resistance to pests, drought, salinity and temperature. RNA interference and other RNA-based approaches can also favor biofuel production pathways in microbes and manufacturing biodegradable plastics to substitute thermoplastics for bioremediation of the environment. In veterinary applications, RNA-based interventions could be used for faster growth of food animals, improved animal breeds, and aquaculture. However, each of these applications has its own regulatory hurdles and biosafety considerations which differ from those of genetically modified organisms. Moreover, scalability and access to RNA technology, particularly in low- and middle-income countries, remain challenging for all applications [Darsan Singh et al 2019, Raybould and Burns 2020, Animasaun and Lawrence 2023].
As artificial intelligence tools become increasingly integrated into RNA therapy development, regulatory agencies must adapt their frameworks to evaluate the safety and efficacy of their use effectively. One primary concern is the need for robust validation processes. Integrating artificial intelligence into RNA therapy development also raises ethical considerations such as data privacy and the potential for bias in algorithm design.
As recommended, pertinent sentences on how structure knowledge influences therapy development have been added throughout the manuscript.
At the beginning and end of section 1 (RNA is an organic code), respectively:
Understanding organic codes is fundamental to therapeutic development.
Therefore, R-loops and interacting proteins are therapeutic targets
At the beginning of section 2 (RNA has many functions through its intricate, ubiquitous, diverse, and dynamic structure):
This has fueled the rise to prominence of RNA-based therapeutics.
To underscore the importance of RNA G-quadruplexes to pathologies and their treatment, we added the following paragraph to section 3 (RNA's structure is defined at primary, secondary, tertiary, quaternary, and quinary levels):
In synucleinopathies, including Parkinson’s disease, dementia with Lewy bodies, and multiple system atrophy, triggered by α-synuclein aggregation leading to progressive neurodegeneration, calcium influx-induced RNA G-quadruplex assembly accelerates α-synuclein phase transition and aggregation, rendering it a therapeutic target [Matsuo et 2024]. To this end, 5-Aminolevulinic acid inhibits the liquid-liquid phase separation of RNA G-quadruplexes, thereby reducing α-synuclein aggregation and associated neurodegeneration [Matsuo et 2024].
In section 4 (long noncoding RNAs):
The therapeutic targeting of lncRNAs has gained traction over the past decade as their diverse functions in gene regulation have been uncovered [194,195]. LncRNAs can modulate chromatin structure, influence transcriptional and post-transcriptional processes, and interact with proteins and other RNAs, making them integral to cellular function [194,195]. For instance, some single-stranded lncRNAs influence chromatin structure by interacting with double- or single-stranded DNA. These interactions can occur as RNA•(DNA)2 triplexes, stabilized by nucleosomes prominently via their histone H3-tail component, or as R-loops. LncRNA-induced R-loops play crucial roles in plants [Gao J et al 2021]. Triplexes follow preferred patterns of palindromic polypyrimidine or polypurine stretches [Buske et al 2011, Statello et al 2021, Maldonado et al 2019, 2023]. An atomic force microscopy-based approach to discriminate triplexes from R-loops could provide single-molecule level insight into lncRNAs’ roles in gene regulation [Merici et al 2024].
The advantages of targeting lncRNAs include their tissue- or cell-type-specific expression patterns [195], acting as scaffolds for protein complexes, enhancers of gene expression, or decoys that inhibit oncogenic pathways [194], and usefulness as diagnostic and prognostic biomarkers [195]. For instance, the MEG3 lncRNA with multiple conserved pseudoknots is involved in the regulatory network of the tumor suppressor p53 [Uroda et al 2019]. Some lncRNAs function as RNA hubs, as illustrated by the long form of NEAT1 [Yamazaki et al 2018], which appears to lack significant self-structure but may participate in long-range interaction between multiple lncRNAs [Lin et al 2018]. Likewise, the lncRNA MALAT1 localizes to nuclear speckles and has many long-range interactions [Lu et al 2016], including with other RNAs such as NEAT1 and U1 snRNA [Cai et al 2020].
For microRNAs (section 4.1.3):
Several miRNAs, including miR-20a, let-7a, miR-17, miR-18a, miR-27a, and miR-92a, give rise to cleavage-inducing tiny RNAs (cityRNAs) when truncated to 14 nucleotides. As exceptionally short guide RNAs, cityRNAs uniquely activate Argonaute proteins, particularly Argonaute 3 (AGO3) while inhibiting AGO2, for target RNA cleavage. This finding has significant implications for understanding RNA interference mechanisms and gene regulation, underlying new possible RNA-based therapeutics and research tools [Park MS et al 2020, Zhang H et al 2024].
We added section 4.5.6 Specificity of action and other challenges
Secondary structures and high similarity within sequence families render specific knockdown of particular noncoding RNAs challenging. Target site specificity can limit the application of several modalities of RNA-based therapeutics. In infectious diseases, there is also the challenge of microbial mutational escape.
We added to section 4.6. Future gene therapy perspectives
The field of RNA therapeutics is rapidly evolving, with emerging technologies poised to enhance the potency and specificity of RNA-based therapies. Innovations such as untranslated region optimization and machine learning-based synthetic RNA motif design are expected to improve therapeutic outcomes significantly. The evolution of artificial intelligence in RNA therapy has progressed from simple predictive models to sophisticated design engines. Artificial intelligence tools are increasingly utilized in RNA therapy for various purposes, including RNA structure prediction as discussed for the Atomic Rotationally Equivariant Scorer (ARES), which has demonstrated high accuracy in predicting RNA structures, surpassing previous methods [139]. For sequence optimization, machine learning approaches are used to analyze large datasets of RNA sequences and their functional outcomes to identify optimal patterns for specific therapeutic applications [Hwang H et al. 2024]. In academia and industry, novel artificial intelligence-based algorithms are being developed to predict tissue-specific regulatory mechanisms of RNA expression and target identification. Among the latter are binding sites of proteins and microRNAs, chemical library screening, hit-to-lead optimization, and personalized RNA therapies tailored to patients' genetic profiles and disease characteristics.
While artificial intelligence tools show promise in accelerating the development of RNA therapeutics, challenges remain. These include the need for high-quality and comprehensive data, especially regarding the biological activity of RNA-targeting small molecules. Noncoding RNA analysis, multi-omics, and meta-transcriptomics data must also be included. Additionally, ensuring the interpretability and transparency of artificial intelligence models used in RNA therapy development is crucial for regulatory approval and building trust among researchers and clinicians.
The future of RNA-based gene therapy may increasingly rely on combination approaches that integrate multiple RNA modalities or combine RNA therapeutics with traditional small molecule drugs and immunotherapies [193]. These strategies could provide synergistic effects, enabling more effective treatment of complex diseases.
Advances in RNA engineering and synthetic biology are expanding the toolkit for gene therapy. For instance, CRISPR-Cas systems using guide RNAs offer unprecedented precision in genome editing, opening new possibilities for treating genetic disorders [194, 199]. The guide RNAs contain hairpin structures that bind to exogenously introduced Cas9 protein and direct it to specific genomic DNA loci for targeted gene editing [Knott and Doudna, 2018]. These technologies enhance the specificity of gene manipulation and allow for targeted interventions to address the underlying causes of diseases.
In an alternative therapeutic approach based on RNA structure, the chimeric degrader concept has been extended to the RNA field as ribonuclease targeting chimeras (RIBOTACs) [Costales et al. 2018]. RIBOTACs use synthetic small molecules that recognize secondary or tertiary RNA structures or antisense oligonucleotides that recognize RNA primary sequence as a "bait" fragment or guide arm. In contrast, the effector arm recruits the endogenous ribonuclease (RNase) L, causing degradation of the target RNA without affecting the host transcriptome [Costales et al., 2018, 2020; Liu X et al. 2020; Meyer et al., 2020].
The highlighted sentences were added to the Conclusions section:
Since the central molecular biology dogma was formulated almost seven decades ago, mounting evidence revealed that RNA is the main determinant of biological diversity. This is driven by RNA’s abundance, modifications, and structural versatility of its coding and noncoding versions, which occasionally overlap. RNA structure-function relationships also vary among cells in an organism, determining cellular identity. Therefore, understanding RNA’s tertiary, quaternary, and quinary structures and their functional relationships remains challenging. The challenge is driven by the ability of RNA to adopt various structures from the secondary to the quinary levels and undergo diverse modifications according to circumstances. This has quelled the impact of traditional computational approaches more than their intrinsic limitations. Advances in approaches, including the use of machine learning and artificial intelligence [139], data gatherings, such as the United States National Institutes of Health’s and National Academies of Sciences’ RNA sequencing initiative including its modifications [209], and novel imaging methods enabling more detailed RNA analysis within single cells and intact tissues [210] provide an optimistic outlook for the continued development and refinement of disruptive RNA-based approaches for medical therapy, diagnosis, and prevention, and agriculture and industrial applications [209].
As artificial intelligence tools become increasingly integrated into RNA therapy development, regulatory agencies must adapt their frameworks to evaluate the safety and efficacy of their use effectively. One primary concern is the need for robust validation processes. Integrating artificial intelligence into RNA therapy development also raises ethical considerations such as data privacy and the potential for bias in algorithm design.
- The quality of Figure 1 requires improvement, as it appears blurry. Adding transcription direction indicators for the RNA secondary structure within the figure would be helpful. Additionally, the color scheme used in Figure 4 seems unconventional and detracts from its impact. Since this figure is central to the manuscript, a more vivid and scientifically accurate representation is strongly recommended.
At the request of another reviewer, we added a new Figure 1 to compare the structures of DNA and RNA. We also changed the old Figure 1 to a new Figure 2 showing the primary, secondary and tertiary structures of a transfer RNA (the choice of RNA was suggested by another reviewer). The 3D structure schematic was obtained from PDB as loaded by the authors that determined the crystal structure for the Escherichia coli tRNA for phenyalalnine. Because we used a tRNA to illustrate primary to tertiary structures, we added new Figure 4 to the quaternary structure section, showing phenylalanine-tRNAPhe bound to the cyclodipeptide synthase RNA-binding protein from Candidatus Glomeribacter gigasporarum. We redid old Figure 4, which is now Figure 5. Figure is a comparison table of viral and non-viral vector-based RNA delivery methods.
Reviewer 3 Report
Comments and Suggestions for Authors
The manuscript titled "RNA structure: Past, future, and gene therapy applications" provides a review of RNA biology, emphasizing its structural and functional complexity. It traces the historical understanding of RNA as a mere intermediary molecule to its current recognition as a versatile biomolecule influencing cellular processes, gene regulation, and phenotypic diversity. The review concludes by exploring perspectives on RNA-based therapeutics and their transformative potential in personalized medicine and regenerative therapies. While the manuscript is well-written, I would like to raise the following points for the authors to address to enhance its novelty and better distinguish it from other excellent reviews on the topic:
Major Comments for the Authors:
- While the manuscript is comprehensive, the sections on RNA structure determination methods would benefit from more detailed comparisons of current computational tools and their limitations. This addition would provide a clearer understanding of the challenges researchers face in RNA structural biology.
- The discussion of RNA structural studies in vivo should be expanded. Although mentioned, the challenges of capturing dynamic RNA conformations within cellular environments could be further elaborated.
- The manuscript mentions advancements in machine learning and AI but does not analyze their specific contributions or limitations in RNA structure prediction or therapy design sufficiently.
- The focus on RNA in gene therapy is robust, but other potential applications, such as in agriculture or environmental sciences, are sparsely covered. Including these areas could broaden the manuscript's appeal and impact.
- The quality and content of figures need significant improvement, as they currently do not add much value to the text. More detailed and visually engaging figures, particularly for RNA structures and gene therapy applications, would enhance understanding. For example, the authors should include tRNAs as an example of RNA tertiary structure in Figure 1.
- Figures 2 and 3 are basic and could be enhanced significantly. The relevant discussions in the text should also be expanded to provide more depth.
- Lines 107–108: Including an additional figure comparing and contrasting DNA and RNA structures would significantly improve the manuscript.
- Section 5, which is particularly interesting, would benefit from additional illustrations and figures to better convey the content.
- Certain concepts, such as RNA's role in gene therapy, are reiterated multiple times across sections without adding new insights. This repetition makes the text appear redundant in places.
- The manuscript largely presents a positive outlook on RNA-based therapeutics but lacks critical analysis of potential pitfalls, such as regulatory hurdles, scalability issues, or long-term safety concerns.
- In light of the above, Section 5.1.1 requires further expansion. The authors should include additional discussion and references to enrich the section.
- Although the manuscript mentions lncRNAs and miRNAs in the context of gene therapy, it does not comprehensively address the broader functional roles of non-coding RNAs (ncRNAs). These could include:
- Regulation of Gene Expression: ncRNAs play crucial roles in epigenetic regulation, transcriptional control, and post-transcriptional modulation.
- Cellular Processes: They are central to processes like RNA splicing, nuclear architecture, and signal transduction.
- Disease Mechanisms: ncRNAs are implicated in various diseases, including cancer, neurological disorders, and cardiovascular conditions, providing insights into their therapeutic potential beyond gene therapy.
Minor Comments:
- In the author contribution section, the authors state, "Both authors (W.A.H., K.H., and R.P.) contributed to writing—original draft preparation, writing—review, and editing." Since there are three authors, the term "both" should be replaced with "all."
- Please correct the format of Reference 132, as it appears inconsistent with the journal's style.
Author Response
Thank you for your time, effort, thorough review, and recommendations for improving the manuscript. Answers to your queries are below; changes are highlighted in the revised manuscript.
- While the manuscript is comprehensive, the sections on RNA structure determination methods would benefit from more detailed comparisons of current computational tools and their limitations. This addition would provide a clearer understanding of the challenges researchers face in RNA structural biology.
We added the following highlighted sections to the paragraph discussing computational tools for structure prediction:
Developed during the last two decades [135], some RNA 3D structure prediction computational tools use high-resolution homologs' more precise structural information to annotate the base-pairing interactions in low-resolution structures in coarse-grained models/simulations [136,137] or in imaging data missing atoms [138]. Moreover, a machine-learning approach, termed Atomic Rotationally Equivariant Scorer (ARES), identifies accurate structural models without assumptions about their defining characteristics despite being trained with the atomic coordinates of only 18 known RNA structures [139]. ARES predicted RNA three-dimensional structures with accuracy surpassing both human expertise and previously established methods.
We added to section 4.6. Future gene therapy perspectives
The field of RNA therapeutics is rapidly evolving, with emerging technologies poised to enhance the potency and specificity of RNA-based therapies. Innovations such as untranslated region optimization and machine learning-based synthetic RNA motif design are expected to improve therapeutic outcomes significantly. The evolution of artificial intelligence in RNA therapy has progressed from simple predictive models to sophisticated design engines. Artificial intelligence tools are increasingly utilized in RNA therapy for various purposes, including RNA structure prediction as discussed for the Atomic Rotationally Equivariant Scorer (ARES), which has demonstrated high accuracy in predicting RNA structures, surpassing previous methods [139]. For sequence optimization, machine learning approaches are used to analyze large datasets of RNA sequences and their functional outcomes to identify optimal patterns for specific therapeutic applications [Hwang H et al. 2024]. In academia and industry, novel artificial intelligence-based algorithms are being developed to predict tissue-specific regulatory mechanisms of RNA expression and target identification. Among the latter are binding sites of proteins and microRNAs, chemical library screening, hit-to-lead optimization, and personalized RNA therapies tailored to patients' genetic profiles and disease characteristics.
While artificial intelligence tools show promise in accelerating the development of RNA therapeutics, challenges remain. These include the need for high-quality and comprehensive data, especially regarding the biological activity of RNA-targeting small molecules. Noncoding RNA analysis, multi-omics, and meta-transcriptomics data must also be included. Additionally, ensuring the interpretability and transparency of artificial intelligence models used in RNA therapy development is crucial for regulatory approval and building trust among researchers and clinicians.
The future of RNA-based gene therapy may increasingly rely on combination approaches that integrate multiple RNA modalities or combine RNA therapeutics with traditional small molecule drugs and immunotherapies [193]. These strategies could provide synergistic effects, enabling more effective treatment of complex diseases.
Advances in RNA engineering and synthetic biology are expanding the toolkit for gene therapy. For instance, CRISPR-Cas systems using guide RNAs offer unprecedented precision in genome editing, opening new possibilities for treating genetic disorders [194, 199]. The guide RNAs contain hairpin structures that bind to exogenously introduced Cas9 protein and direct it to specific genomic DNA loci for targeted gene editing [Knott and Doudna, 2018]. These technologies enhance the specificity of gene manipulation and allow for targeted interventions to address the underlying causes of diseases.
In an alternative therapeutic approach based on RNA structure, the chimeric degrader concept has been extended to the RNA field as ribonuclease targeting chimeras (RIBOTACs) [Costales et al. 2018]. RIBOTACs use synthetic small molecules that recognize secondary or tertiary RNA structures or antisense oligonucleotides that recognize RNA primary sequence as a "bait" fragment or guide arm. In contrast, the effector arm recruits the endogenous ribonuclease (RNase) L, causing degradation of the target RNA without affecting the host transcriptome [Costales et al., 2018, 2020; Liu X et al. 2020; Meyer et al., 2020].
The highlighted sentences were added to the Conclusions section:
Since the central molecular biology dogma was formulated almost seven decades ago, mounting evidence revealed that RNA is the main determinant of biological diversity. This is driven by RNA’s abundance, modifications, and structural versatility of its coding and noncoding versions, which occasionally overlap. RNA structure-function relationships also vary among cells in an organism, determining cellular identity. Therefore, understanding RNA’s tertiary, quaternary, and quinary structures and their functional relationships remains challenging. The challenge is driven by the ability of RNA to adopt various structures from the secondary to the quinary levels and undergo diverse modifications according to circumstances. This has quelled the impact of traditional computational approaches more than their intrinsic limitations. Advances in approaches, including the use of machine learning and artificial intelligence [139], data gatherings, such as the United States National Institutes of Health’s and National Academies of Sciences’ RNA sequencing initiative including its modifications [209], and novel imaging methods enabling more detailed RNA analysis within single cells and intact tissues [210] provide an optimistic outlook for the continued development and refinement of disruptive RNA-based approaches for medical therapy, diagnosis, and prevention, and agriculture and industrial applications [209].
As artificial intelligence tools become increasingly integrated into RNA therapy development, regulatory agencies must adapt their frameworks to evaluate the safety and efficacy of their use effectively. One primary concern is the need for robust validation processes. Integrating artificial intelligence into RNA therapy development also raises ethical considerations such as data privacy and the potential for bias in algorithm design.
Please also see answer to question 3.
- The discussion of RNA structural studies in vivo should be expanded. Although mentioned, the challenges of capturing dynamic RNA conformations within cellular environments could be further elaborated.
We added the following sentences to section 2:
To this end, the sc-SPORT high-throughput approach allows the study of RNA structures in single cells by optimizing conditions that increase mutation rates and efficiencies of library preparation and second-strand synthesis. Although the approach also includes a computational pipeline to analyze heterogeneous RNA structures and identify them transcriptome-wide, it captures only a few hundred cells (>300) in one experiment, which can be overcome with future modifications [82]. To this end, the sc-SPORT high-throughput approach allows the study of RNA structures in single cells by optimizing conditions that increase mutation rates and efficiencies of library preparation and second-strand synthesis. Although the approach also includes a computational pipeline to analyze heterogeneous RNA structures and identify them transcriptome-wide, it captures only a few hundred cells (>300) in one experiment, which can be overcome with future modifications [82]. Moreover, combining long- and short-read sequencing in single cells and isolated nuclei, has revealed new messenger RNAs, approximately three-fourths of the brain transcriptome, in neurodegenerative diseases, such as Alzheimer’s disease, dementia with Lewy bodies, or Parkinson’s disease [Lui CS et al 2024].
In section 4, under long noncoding RNAs:
The therapeutic targeting of lncRNAs has gained traction over the past decade as their diverse functions in gene regulation have been uncovered [194,195]. LncRNAs can modulate chromatin structure, influence transcriptional and post-transcriptional processes, and interact with proteins and other RNAs, making them integral to cellular function [194,195]. For instance, some single-stranded lncRNAs influence chromatin structure by interacting with double- or single-stranded DNA. These interactions can occur as RNA•(DNA)2 triplexes, stabilized by nucleosomes prominently via their histone H3-tail component, or as R-loops. LncRNA-induced R-loops play crucial roles in plants [Gao J et al 2021]. Triplexes follow preferred patterns of palindromic polypyrimidine or polypurine stretches [Buske et al 2011, Statello et al 2021, Maldonado et al 2019, 2023]. An atomic force microscopy-based approach to discriminate triplexes from R-loops could provide single-molecule level insight into lncRNAs’ roles in gene regulation [Merici et al 2024].
The advantages of targeting lncRNAs include their tissue- or cell-type-specific expression patterns [195], acting as scaffolds for protein complexes, enhancers of gene expression, or decoys that inhibit oncogenic pathways [194], and usefulness as diagnostic and prognostic biomarkers [195]. For instance, the MEG3 lncRNA with multiple conserved pseudoknots is involved in the regulatory network of the tumor suppressor p53 [Uroda et al 2019]. Some lncRNAs function as RNA hubs, as illustrated by the long form of NEAT1 [Yamazaki et al 2018], which appears to lack significant self-structure but may participate in long-range interaction between multiple lncRNAs [Lin et al 2018]. Likewise, the lncRNA MALAT1 localizes to nuclear speckles and has many long-range interactions [Lu et al 2016], including with other RNAs such as NEAT1 and U1 snRNA [Cai et al 2020].
- The manuscript mentions advancements in machine learning and AI but does not analyze their specific contributions or limitations in RNA structure prediction or therapy design sufficiently.
The following sentences were added to the section preceding the conclusion.
The evolution of artificial intelligence in RNA therapy has progressed from simple predictive models to sophisticated design engines. Artificial intelligence tools are increasingly utilized in RNA therapy for various purposes, including RNA structure prediction as discussed for the Atomic Rotationally Equivariant Scorer (ARES), which has demonstrated high accuracy in predicting RNA structures, surpassing previous methods [139]. For sequence optimization, machine learning approaches are used to analyze large datasets of RNA sequences and their functional outcomes to identify optimal patterns for specific therapeutic applications [Hwang H et al. 2024]. In academia and industry, novel artificial intelligence-based algorithms are being developed to predict tissue-specific regulatory mechanisms of RNA expression and target identification. Among the latter are binding sites of proteins and microRNAs, chemical library screening, hit-to-lead optimization, and personalized RNA therapies tailored to patients' genetic profiles and disease characteristics.
While artificial intelligence tools show promise in accelerating the development of RNA therapeutics, challenges remain. These include the need for high-quality and comprehensive data, especially regarding the biological activity of RNA-targeting small molecules. Noncoding RNA analysis, multi-omics, and meta-transcriptomics data must also be included. Additionally, ensuring the interpretability and transparency of artificial intelligence models used in RNA therapy development is crucial for regulatory approval and building trust among researchers and clinicians.
The highlighted sentences were added to the Conclusions section:
Since the central molecular biology dogma was formulated almost seven decades ago, mounting evidence revealed that RNA is the main determinant of biological diversity. This is driven by RNA’s abundance, modifications, and structural versatility of its coding and noncoding versions, which occasionally overlap. RNA structure-function relationships also vary among cells in an organism, determining cellular identity. Therefore, understanding RNA’s tertiary, quaternary, and quinary structures and their functional relationships remains challenging. The challenge is driven by the ability of RNA to adopt various structures from the secondary to the quinary levels and undergo diverse modifications according to circumstances. This has quelled the impact of traditional computational approaches more than their intrinsic limitations. Advances in approaches, including the use of machine learning and artificial intelligence [139], data gatherings, such as the United States National Institutes of Health’s and National Academies of Sciences’ RNA sequencing initiative including its modifications [209], and novel imaging methods enabling more detailed RNA analysis within single cells and intact tissues [210] provide an optimistic outlook for the continued development and refinement of disruptive RNA-based approaches for medical therapy, diagnosis, and prevention, and agriculture and industrial applications [209].
As artificial intelligence tools become increasingly integrated into RNA therapy development, regulatory agencies must adapt their frameworks to evaluate the safety and efficacy of their use effectively. One primary concern is the need for robust validation processes. Integrating artificial intelligence into RNA therapy development also raises ethical considerations such as data privacy and the potential for bias in algorithm design.
- The focus on RNA in gene therapy is robust, but other potential applications, such as in agriculture or environmental sciences, are sparsely covered. Including these areas could broaden the manuscript's appeal and impact.
We added the following paragraph to the Conclusions section:
Beyond the medical uses of RNA-based therapies, which is the scope of the present review, agricultural applications include silencing RNAs and messenger RNA modifications for crop yield enhancement, RNA interference to introduce favorable traits, and RNA technologies to increase resistance to pests, drought, salinity and temperature. RNA interference and other RNA-based approaches can also favor biofuel production pathways in microbes and manufacturing biodegradable plastics to substitute thermoplastics for bioremediation of the environment. In veterinary applications, RNA-based interventions could be used for faster growth of food animals, improved animal breeds, and aquaculture. However, each of these applications has its own regulatory hurdles and biosafety considerations as they differ from those of genetically modified organisms. Moreover, accessing RNA technology in low- and middle-income countries remains challenging for all applications [Darsan Singh et al 2019, Raybould and Burns 2020, Animasaun and Lawrence 2023].
- The quality and content of figures need significant improvement, as they currently do not add much value to the text. More detailed and visually engaging figures, particularly for RNA structures and gene therapy applications, would enhance understanding. For example, the authors should include tRNAs as an example of RNA tertiary structure in Figure 1.
All previous figures have been modified. As another reviewer requested, there is a new Figure 1 comparing the structures of DNA and RNA. The old Figure 1 is the new Figure 2. As recommended, we used the Escherichia coli tRNA for phenylalanine to illustrate primary to tertiary structure. We blended old Figures 3 and 4 into a new Figure 3, to which we added the hammerhead ribozyme structure containing a pseudoknot. Because we used a tRNA to illustrate primary to tertiary structures, we added Figure 4 to the quaternary structure section to show the same tRNA bound to the cyclodipeptide synthase RNA-binding protein from Candidatus Glomeribacter gigasporarum. The old Figure 4 was redrawn and is now Figure 5. The new figure 4 shows a comparison between viral and non-rival-based RNA delivery methods.
- Figures 2 and 3 are basic and could be enhanced significantly. The relevant discussions in the text should also be expanded to provide more depth.
Old figures 2 and 3 were merged into new Figure 3, which now includes the hammerhead ribozyme containing a pseudoknot.
The following sentences were added to the text to provide more depth to the section alluding to new Figure 3:
Ribozymes are RNA enzymes that catalyze essential cellular reactions and are potential therapeutic agents. Nucleolytic ribozymes catalyze phosphoryl transfer reactions. The nucleolytic hammerhead ribozyme containing a pseudoknot, shown in Figure 3B, is a widespread example, including within the human genome [Zhan X et al 2024]. Catalytic RNA molecules possess simultaneously a genotype and a phenotype, bypassing protein expression as the determinant of phenotype as per the central molecular biology dogma. Thanks to differential folding and catalytic activity, a single RNA genotype has the potential to adopt two or perhaps more distinct phenotypes [Vaidya and Lehman 2009].
In synucleinopathies, including Parkinson’s disease, dementia with Lewy bodies, and multiple system atrophy, triggered by α-synuclein aggregation leading to progressive neurodegeneration, calcium influx-induced RNA G-quadruplex assembly accelerates α-synuclein phase transition and aggregation, rendering it a therapeutic target [Matsuo et 2024]. To this end, 5-Aminolevulinic acid inhibits the liquid-liquid phase separation of RNA G-quadruplexes, thereby reducing α-synuclein aggregation and associated neurodegeneration [Matsuo et 2024].
- Lines 107–108: Including an additional figure comparing and contrasting DNA and RNA structures would significantly improve the manuscript.
This figure has been added as the new Figure 1.
- Section 5, which is particularly interesting, would benefit from additional illustrations and figures to better convey the content.
Figure 4 was added comparing the viral and non-viral- based RNA delivery methods. The last Figure, now Figure 5, was reformatted and more details were added.
- Certain concepts, such as RNA's role in gene therapy, are reiterated multiple times across sections without adding new insights. This repetition makes the text appear redundant in places.
We have deleted these redundancies.
- The manuscript largely presents a positive outlook on RNA-based therapeutics but lacks critical analysis of potential pitfalls, such as regulatory hurdles, scalability issues, or long-term safety concerns.
We added the following to the Conclusions section:
Beyond the medical uses of RNA-based therapies, which is the scope of the present review, agricultural applications include silencing RNAs and messenger RNA modifications for crop yield enhancement, RNA interference to introduce favorable traits, and RNA technologies to increase resistance to pests, drought, salinity and temperature. RNA interference and other RNA-based approaches can also favor biofuel production pathways in microbes and manufacturing biodegradable plastics to substitute thermoplastics for bioremediation of the environment. In veterinary applications, RNA-based interventions could be used for faster growth of food animals, improved animal breeds, and aquaculture. However, each of these applications has its own regulatory hurdles and biosafety considerations which differ from those of genetically modified organisms. Moreover, scalability and access to RNA technology, particularly in low- and middle-income countries, remain challenging for all applications [Darsan Singh et al 2019, Raybould and Burns 2020, Animasaun and Lawrence 2023].
As artificial intelligence tools become increasingly integrated into RNA therapy development, regulatory agencies must adapt their frameworks to evaluate the safety and efficacy of their use effectively. One primary concern is the need for robust validation processes. Integrating artificial intelligence into RNA therapy development also raises ethical considerations such as data privacy and the potential for bias in algorithm design
The following was added to the section preceding the Conclusions section:
While artificial intelligence tools show promise in accelerating the development of RNA therapeutics, challenges remain. These include the need for high-quality and comprehensive data, especially regarding the biological activity of RNA-targeting small molecules. Noncoding RNA analysis, multi-omics, and meta-transcriptomics data must also be included. Additionally, ensuring the interpretability and transparency of artificial intelligence models used in RNA therapy development is crucial for regulatory approval and building trust among researchers and clinicians.
- In light of the above, Section 5.1.1 requires further expansion. The authors should include additional discussion and references to enrich the section.
Section 5.1.1 is now section 4.1.1., to which the highlighted paragraphs were added.
mRNA is a powerful tool in gene therapy, particularly for protein replacement and vaccination strategies [184]. This approach directly delivers synthetic mRNA-encoding therapeutic proteins into cells [185]. Once there, cellular machinery translates it into functional proteins [184]. The rapid development and success of mRNA vaccines against SARS-CoV-2 have showcased the potential of this technology, accelerating research into mRNA therapeutics for other diseases [186].
The therapeutic use of mRNA offers several advantages, including transient expression without genomic integration, which enhances safety [184]; the ability to produce almost any functional protein or peptide in the human body [185,186]; faster design and production compared to conventional approaches [185]; cost-effectiveness and flexibility [185]; and higher transfection efficiency and lower toxicity compared to DNA-based drugs [185,186].
Transient expression without genomic integration enhances safety in mRNA-based therapies. Unlike DNA-based approaches, mRNA does not need to enter the nucleus to function, eliminating the risk of insertional mutagenesis [Jahanafrooz et al., 2020]. This transient nature ensures that the mRNA is only active for a limited time, reducing the potential burden on host homeostasis and decreasing off-target effects in applications such as gene editing [Karim et al., 2022].
One of the main advantages of mRNA therapeutics is their ability to produce nearly any functional protein or peptide in the human body. This versatility paves the way for treatments targeting various diseases, including those previously deemed unmanageable or genetic [Sahin et al., 2014; Qin et al., 2022; Youssef et al., 2023]. The programmable aspect of mRNA allows for the rapid production of various proteins, making it a powerful tool for precision medicine [Qin et al., 2022].
Additionally, mRNA therapeutics offer faster design and production times than traditional methods [Youssef et al., 2023]. mRNA's simplicity makes it well-suited for studying short-term genetic effects and for the quick creation of recombinant proteins. This advantage is significant in responding to large-scale outbreaks of infectious diseases, as evidenced by the rapid development of mRNA vaccines for COVID-19 [Chaudhary et al., 2021; Szabó et al., 2022].
Cost-effectiveness and flexibility are key factors that enhance the attractiveness of mRNA therapeutics [Duan et al., 2022]. Generally, the production and manufacturing of mRNA are less expensive and more convenient than proteins, especially when creating vaccine products during a pandemic [Light & Lexchin, 2021; Gote et al., 2023].
Additionally, mRNA therapeutics provide greater transfection efficiency and lower toxicity when compared to DNA-based drugs [Juncker et al., 2023; Zhao et al., 2006]. Research shows that mRNA electroporation can achieve up to 98% transfection efficiency across various cell lines, significantly outperforming DNA-based systems [Juncker et al., 2023]. Furthermore, mRNA transfection generally results in higher cell viability than DNA transfection, making it a safer option for cellular manipulation [Juncker et al., 2023].
The history of mRNA therapeutics dates back to the early 1960s when mRNA was discovered as a critical player in genetic information flow [184,185]. However, it was in the late 1980s that researchers began exploring mRNA as a therapeutic tool [185]. In 1987, Robert Malone from the Salk Institute demonstrated that synthetic mRNA strands mixed with lipid particles could transfect human cells to express proteins of interest [185].
As the field continues to evolve, over 54 mRNA vaccines and drugs are currently in various stages of clinical testing for multiple diseases, from infectious to cardiovascular conditions [185]. The versatility and rapid production capabilities of mRNA therapeutics position them as a promising tool in the future of precision medicine.
- Although the manuscript mentions lncRNAs and miRNAs in the context of gene therapy, it does not comprehensively address the broader functional roles of non-coding RNAs (ncRNAs). These could include:
- Regulation of Gene Expression: ncRNAs play crucial roles in epigenetic regulation, transcriptional control, and post-transcriptional modulation.
- Cellular Processes: They are central to processes like RNA splicing, nuclear architecture, and signal transduction.
- Disease Mechanisms: ncRNAs are implicated in various diseases, including cancer, neurological disorders, and cardiovascular conditions, providing insights into their therapeutic potential beyond gene therapy.
A section on non-coding RNAs (4.1.4) was added, of which long-noncoding RNAs became a subsection (4.1.4.1)
- Non-coding RNAs
Non-coding RNAs (ncRNAs) are pivotal in regulating gene expression and cellular processes, making them essential components in the gene therapy landscape. These molecules are not merely transcriptional byproducts but integral to various biological functions, influencing epigenetic regulation, transcriptional control, and post-transcriptional modulation of gene expression [Kaikkonen et al., 2011; Statello et al., 2021; Patil et al., 2013; Gao et al., 2020; Bu et al., 2023].
In addition to their regulatory functions, ncRNAs are central to numerous cellular processes such as RNA splicing, nuclear architecture maintenance, and signal transduction pathways [Statello et al., 2021; Chen et al., 2022]. One of the key processes in which ncRNAs are involved is RNA splicing, which contributes to the precise removal of introns and the joining of exons in pre-mRNA transcripts [Razin & Gavrilov, 2021; Frías-Lasserre & Villagra, 2017]. This splicing is crucial for generating mature mRNA molecules that encode functional proteins, influencing gene expression outcomes.
ncRNAs are integral to maintaining nuclear architecture, as they help organize chromatin and facilitate interactions between different genomic regions [Razin & Gavrilov, 2021]. This spatial organization is essential for proper gene expression and cellular function. Furthermore, ncRNAs participate in various signal transduction pathways, mediating extracellular signals to intracellular responses [Zong et al., 2023; Ramón y Cajal et al., 2019; Di et al., 2023]. By influencing these pathways, ncRNAs can modulate critical cellular processes such as metabolism, immune responses, and stress responses.
ncRNAs regulate other signaling pathways that govern essential cellular functions such as proliferation, differentiation, and apoptosis [Statello et al., 2021]. For instance, specific microRNAs (miRNAs) can target mRNAs encoding proteins involved in cell cycle regulation, thereby promoting or inhibiting cell division [Liang & He, 2011; Sati & Parhar, 2021]. This regulatory capacity is vital for maintaining tissue homeostasis and ensuring cells respond appropriately to developmental cues and environmental stimuli.
ncRNAs play a crucial role in apoptosis by modulating the expression of pro-apoptotic and anti-apoptotic factors [Ghafouri-Fard et al., 2021]. This determines whether a cell will undergo programmed cell death or survive under stress conditions. The delicate balance ncRNAs maintain is fundamental to an organism's overall health [Ghafouri-Fard et al., 2021].
The involvement of ncRNAs in disease mechanisms has garnered significant attention due to their implications in various pathologies, including cancer, neurological disorders, and cardiovascular diseases [Nemeth et al., 2023; Beňačka et al., 2023]. Furthermore, ncRNAs have been shown to regulate processes such as cardiac hypertrophy and vascular remodeling in cardiovascular diseases [Viereck et al., 2020; Kawaguchi et al., 2023]. Understanding the roles of ncRNAs in these diseases provides insights into their underlying mechanisms. It opens new avenues for developing targeted therapies that harness the unique properties of these regulatory molecules.
- Long non-coding RNAs
Long non-coding RNAs (lncRNAs) are becoming significant targets and tools in gene therapy, and their potential applications are expanding rapidly [194]. Typically longer than 200 nucleotides, lncRNAs play crucial roles in gene regulation and various cellular processes, influencing complex genetic networks [194,195]. Manipulating them through gene therapy may offer new avenues for treating complex genetic disorders and cancers.
The therapeutic targeting of lncRNAs has gained traction over the past decade as their diverse functions in gene regulation have been uncovered [194,195]. LncRNAs can modulate chromatin structure, influence transcriptional and post-transcriptional processes, and interact with proteins and other RNAs, making them integral to cellular function [194,195]. For instance, some single-stranded lncRNAs influence chromatin structure by interacting with double- or single-stranded DNA. These interactions can occur as RNA•(DNA)2 triplexes, stabilized by nucleosomes prominently via their histone H3-tail component, or as R-loops. LncRNA-induced R-loops play crucial roles in plants [Gao J et al 2021]. Triplexes follow preferred patterns of palindromic polypyrimidine or polypurine stretches [Buske et al 2011, Statello et al 2021, Maldonado et al 2019, 2023]. An atomic force microscopy-based approach to discriminate triplexes from R-loops could provide single-molecule level insight into lncRNAs’ roles in gene regulation [Merici et al 2024].
The advantages of targeting lncRNAs include their tissue- or cell-type-specific expression patterns [195], acting as scaffolds for protein complexes, enhancers of gene expression, or decoys that inhibit oncogenic pathways [194], and usefulness as diagnostic and prognostic biomarkers [195]. For instance, the MEG3 lncRNA with multiple conserved pseudoknots is involved in the regulatory network of the tumor suppressor p53 [Uroda et al 2019]. Some lncRNAs function as RNA hubs, as illustrated by the long form of NEAT1 [Yamazaki et al 2018], which appears to lack significant self-structure but may participate in long-range interaction between multiple lncRNAs [Lin et al 2018]. Likewise, the lncRNA MALAT1 localizes to nuclear speckles and has many long-range interactions [Lu et al 2016], including with other RNAs such as NEAT1 and U1 snRNA [Cai et al 2020].
Ongoing research is exploring various strategies for effectively targeting lncRNAs. These strategies include transcriptional inhibition, post-transcriptional modulation, and using CRISPR technology to edit lncRNA expression patterns or genomic loci [194,195].
LncRNAs are involved in numerous diseases. For instance, studies have shown that targeting specific oncogenic lncRNAs can inhibit tumor growth and metastasis in preclinical models [194]. Furthermore, restoring downregulated or lost lncRNAs presents an exciting opportunity for therapeutic development [194].
For microRNAs (section 4.1.3):
Several miRNAs, including miR-20a, let-7a, miR-17, miR-18a, miR-27a, and miR-92a, give rise to cleavage-inducing tiny RNAs (cityRNAs) when truncated to 14 nucleotides. As exceptionally short guide RNAs, cityRNAs uniquely activate Argonaute proteins, particularly Argonaute 3 (AGO3) while inhibiting AGO2, for target RNA cleavage. This finding has significant implications for understanding RNA interference mechanisms and gene regulation, underlying new possible RNA-based therapeutics and research tools [Park MS et al 2020, Zhang H et al 2024].
Minor Comments:
- In the author contribution section, the authors state, "Both authors (W.A.H., K.H., and R.P.) contributed to writing—original draft preparation, writing—review, and editing." Since there are three authors, the term "both" should be replaced with "all."
This has been corrected.
- Please correct the format of Reference 132, as it appears inconsistent with the journal's style.
This has been corrected.
Round 2
Reviewer 1 Report
Comments and Suggestions for Authors
The authors have corrected the manuscript carefully.